# Are maps of nitrate reduction in groundwater altered by climate and land use changes?

Ida Karlsson Seidenfaden[1], Torben Obel Sonnenborg[1], Jens Christian Refsgaard[1], Christen Duus Børgesen[2], Jørgen Eivind Olesen[2], Dennis Trolle[3]

[1]Geological Survey of Denmark and Greenland, GEUS, Copenhagen, 1350, Denmark
[2]Aarhus University, Department of Agroecology, Tjele, 8830, Denmark
[3]Aarhus University, Department of Bioscience - Lake Ecology, Silkeborg, 8600, Denmark

*Correspondence to*: Ida K. Seidenfaden, ika@geus.dk

**Abstract.** Nitrate reduction maps have been used routinely in Northern Europe for calculating efficiency of remediation measures and impact of climate change on nitrate leaching. These maps are therefore valuable tools for policy analysis and mitigation targeting. Nitrate reduction maps are normally based on output from complex hydrological models, and once generated, are largely assumed constant in time. However, the distribution, magnitude and efficiency of nitrate reduction can not necessarily be considered stationary during changing climate and land use as flow paths, nitrate release timing and their interaction may shift. This study investigates the potential improvement of using transient nitrate reduction maps compared to a constant nitrate reduction map is assumed during land use and climate change, both for nitrate loads and the spatial variation in reduction. For this purpose, a crop and soil model (DAISY) was setup up to provide nitrate input to a distributed hydrological model (MIKE SHE) for an agricultural catchment in Funen, Denmark. Nitrate reduction maps based on an observed dataset of land use and climate were generated and compared to nitrate reduction maps generated for all combinations of four potential land use change scenarios and four future climate model projections. Nitrate reduction maps were found to be more sensitive to changes in climate, leading to reduction map change of up to 10%; while land use changes effects were minor. The study, however, also showed that the reductions maps are products of a range of complex interactions between water fluxes, nitrate use and timing. What is also important to note, is that the choices made for future scenarios, model setup and assumptions may affect the resulting span in reduction capability. To account for this uncertainty multiple approaches, assumptions and models could be applied for the same area, however as these models are very time consuming this is not always a feasible approach in practice. An uncertainty in the order of 10% on the reduction map may have major impacts on practical water management. It is therefore important to acknowledge if such errors are deemed acceptable in relation to the purpose and context of specific water management situations.

**1 Introduction**

Nitrate loads from agricultural areas are recognized to cause harmful impacts on groundwater and surface water resources, including eutrophication in aquatic ecosystems (Diaz and Rosenberg, 2008). This is also the case in the Baltic Sea drainage basin (Reusch et al., 2018), including Denmark, where nitrate load from agriculture constitutes one of the major water resources management challenges. Nitrate is removed by a set of natural near-surface removal processes including plant uptake and soil retention, furthermore, the natural removal of nitrate in the groundwater and the surface water must also be considered, when

assessing the impacts of nitrate leaching from agricultural areas on aquatic ecosystems. This removal, takes place via natural biogeochemical reduction processes often referred to as denitrification It can be expressed as a percentage removal and depending on the actual hydrobiogeochemical conditions, the denitrification may mainly occur in groundwater or in surface water systems such as lakes or wetlands (Huno et al., 2018; Quick et al., 2019).

In the groundwater zone, nitrate reduction takes place when nitrate containing water migrates from aerobic to anaerobic conditions and inherent reduced compounds are available (Hansen et al., 2014a; Postma et al., 1991). For Quaternary sediments these reduced compounds are mainly organic carbon and pyrite and ferrous ion from clay minerals (Ernstsen and Mørup, 1992; Postma et al., 1991). This transition zone between aerobic and anaerobic conditions is denoted the redox interface. The amount of nitrate reduction occurring in groundwater will then depend on the flow paths and the depth to the redox interface. In areas

with Quaternary sediments characterized by groundwater dominated flow patterns and a relatively shallow redox interface, the nitrate reduction in groundwater can be the dominant removal process (Hansen et al., 2009). For example, Højberg et al. (2015a) estimated that on average 63% of the nitrate leaching in Denmark is removed by nitrate reduction in groundwater.

Heterogeneities in geology and drainage systems are responsible for substantial local spatial variations in nitrate reduction.

However, the spatial variation of nitrate reduction in the groundwater system has so far only been investigated in a handful of studies (e.g. Højberg et al., 2015a; Knoll et al., 2020; Kunkel et al., 2008; Merz et al., 2009; Tesoriero et al., 2015; Wriedt and Rode, 2006). Different approaches have been used in these studies from nitrate groundwater modelling (Højberg et al., 2015b; Merz et al., 2009; Wriedt and Rode, 2006), data driven machine learning (Knoll et al., 2020) or statistical modelling (Tesoriero

et al., 2015). An approach for utilizing and illustrating the results and the spatially varying nitrate removal fractions

(percentages) is through a nitrate reduction map (Hansen et al., 2014a). A nitrate reduction map is typically produced by using

a complex hydrological model, including simulations of root zone nitrate leaching as well as groundwater and surface flow

and transport; and has been applied in several catchments in Denmark and in catchments surrounding the Baltic Sea region

(Hansen et al., 2014b; Højberg et al., 2017; Wulff et al., 2014). Hansen et al. (2014a) produced nitrate reduction maps for a

101 $km^2$ catchment in Denmark, showing that nitrate reduction may vary from 20% to 70% between neighbouring agricultural

fields located only a couple of hundred meters apart. Similarly, Højberg et al. (2015a) and Andersen et al. (2016) estimated

very large variations in nitrate reduction between different regions in Denmark and in the Baltic Sea drainage basin,

respectively.

The efficiencies of remediation measures at different locations on nitrate loadings can easily be calculated with a nitrate

reduction map, and the measures can be spatially targeted to locations, where the natural removal is relatively small and the

mitigation effect hence is relatively large (Hansen et al., 2017; Refsgaard et al., 2019). Similarly, a nitrate reduction map can

be used to transform climate change and other land use changes impacts on nitrate leaching from agricultural areas to a

catchment response (Olesen et al., 2019). Using nitrate reduction maps based on a single model run, is clearly a much faster

method than running multiple complex hydrological simulation models for large ensembles of scenarios and is therefore a

practical tool for policy analysis (Andersen et al., 2016; Højberg et al., 2015a). A severe problem in this respect is, however,

that the nitrate reduction maps may  not be constant in time as the reduction taking place at a given location  depend on resulting

flow pathways (Hansen et al., 2014b). It is therefore very relevant to investigate the potential error arising when nitrate

reduction maps are assumed to be constant in time. No studies have been reported on that issue. Even as, the link between

climate change, land use change and nitrate reduction has been established in previous studies (e.g. Fleck et al., 2017; Mas-

Pla and Menció, 2019; Olesen et al., 2019; Ortmeyer et al., 2021; Sjøeng et al., 2009). Ortmeyer et al. (2021) used a water

balance model combined with a lumped-parameter nitrate mass model for an area in Germany, finding that nitrate

concentrations in the groundwater increased towards the end of the century by up to 89 % as a result of changes in temperature,

evapotranspiration and precipitation. Mas-Pla and Menció (2019) found that climate change in turn affects groundwater

recharge and thus the dilution of nitrate in the subsurface in a study in Catalonia. While, Paradis et al. (2016) found that new

agricultural practices under changing climate conditions led to substantial nitrate increases on an Island in eastern Canada.

The objectives of the present study are to assess i) how nitrate reduction maps showing spatially varying nitrate removal

fractions in the groundwater zone are affected by changes in climate and land use, and ii) the errors in nitrate loading made by

assuming nitrate reduction maps to be constant. The analyses are performed using a complex hydrological simulation model

for a Danish catchment to calculate nitrate reduction maps for the present conditions as well as for scenarios of climate and

land use change. The reduction in this catchment has previously been shown to be dominated by saturated zone reduction

processes (Hansen et al., 2009).

## 2 Study site

The study site is located in the central part of Denmark on the Island of Funen. It consists of the 486 km$^2$ upstream part of the

Odense River basin, where the Kratholm discharge station marks the outlet (Figure 1). The catchment is drained by a 200 km

river network with the outlet located at the Odense Fjord to the northeast. Land use in the area is predominantly agricultural

(68%), mainly pig farms followed by dairy and plant production farms (Figure 1); forest constitutes only 5 %; urban areas are

8%; 1% is water bodies and the remaining are either fallow or grasslands (Nielsen et al., 2000). The soil map and parameters

consists of 10 soil types that are created by Børgesen et al. (2013) and Greve et al. (2007) based on national databases. The

soil type is dominated by clayey soils (71%) with smaller areas of sand (see Karlsson et al. (2015) and Karlsson et al. (2016)

for more information), as a result the agricultural area is heavily drained. The geology is mainly a result of previous glaciations

like till deposits. Aquifers are generally confined, and the phreatic groundwater tables are shallow. The discharge station at

Kratholm (ST45.21) has one of the best nutrient time series in Denmark starting in the 1980s, with near-daily sampling from

1989 (Windolf et al., 2016). The station, therefore, provides a long and near-complete data set for nutrient modelling as well

as an extensive water discharge time series (Trolle et al., 2019). In 2005-2009, the average discharge amounts to 4.6m$^3$/s and

the transport in the stream (load) is approximately 14 kg NO3-N/ha/year, calculated from measurements of mean concentration

and mean water discharge. A decreasing trend in nitrate loads has been observed previously during 2000-2013 by Windolf et

al. (2016), possibly due to implementation of mitigation measures in the catchment. Three other discharge stations are also

present in the catchment (from downstream: ST45.01, ST45.28, ST45.20)

Measurements of the redox depth are available from 226 boreholes in the area, the redox depths were mainly interpretated based on sediment colour as described by e.g. Ernstsen and Mørup (1992), and a few by measurements of reduced compounds. They show a shallow redox interface, where 50% of the measurements have a redox depth less than 4.5 m below terrain, while

90 % of the depths are located in the upper 12.9 meter. An old redox depth map with 1 km resolution, based on measurements and geological interpretation, shows depths between 1 to 5 meters at many locations and between 5 to 15 meters in other locations (Ernstsen et al., 2006). A recent redox depth map of Denmark was created in 2019, where measurements and system variables were used in a machine learning environment to create a detailed redox depth map at 100 meter resolution. This newer map also indicates that the redox depth in the study area is predominantly shallow with 1-10 meter depth, and very few

sites of 10-15 meters depth (Koch et al., 2019a; Koch et al., 2019b).

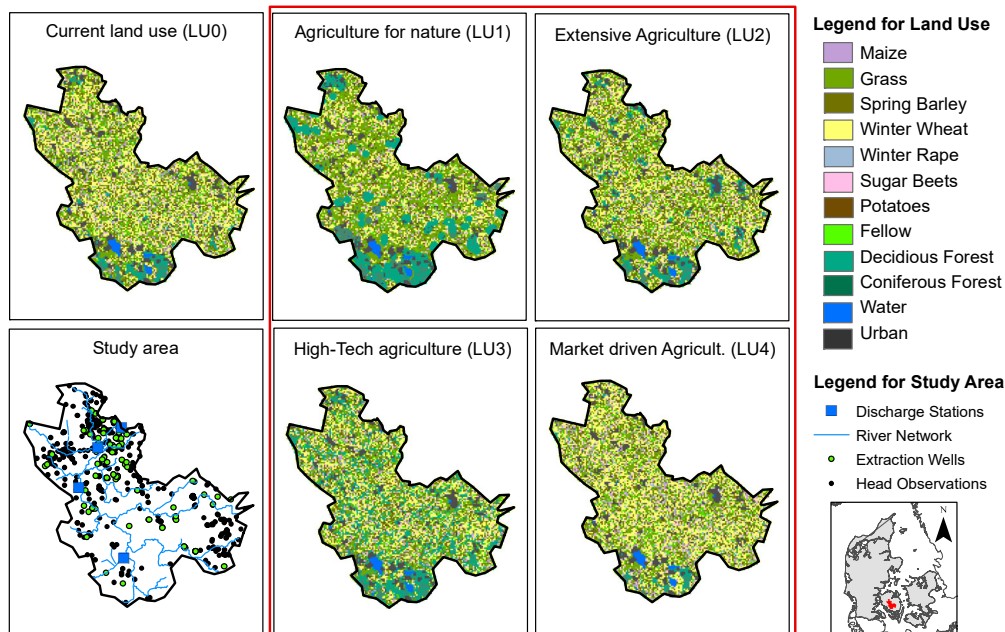

**Figure 1: The current land use distribution (LU0) in the study area (Upper left panel) and the river network and location of observations and extraction wells (Lower left panel). The red square (right and centre) marks the four projected scenarios for land use scenarios LU1-4.**

## 3 Methods

The DAISY-MIKE SHE modelling system is used to describe nitrate transport in the catchment. DAISY is used to quantify leaching of nitrate from the root zone while MIKE SHE is used to simulate the transport and degradation of nitrate in the saturated zone. Both models are forced by the same daily inputs of precipitation, temperature and potential evapotranspiration. Both MIKE SHE and DAISY have been used extensively in the Danish area, and MIKE SHE forms the basis of the national nitrate and groundwater model (Bruun et al., 2003; Hoang et al., 2010; Højberg et al., 2015a; Højberg et al., 2010; Højberg et al., 2013; Højberg et al., 2015b; Troldborg et al., 2010). MIKE SHE is a fully coupled integrated groundwater-surface water model and this integration is important for assessing the feedback between unsaturated and saturated zone, especially under changing climate. However, MIKE SHE does not simulate crops development and nitrate leaching from the root zone, and therefore information from an agrological model, like DAISY, is necessary.

Calibration is done in three phases: 1. The MIKE SHE model is automatically calibrated using discharge and hydraulic heads (section 3.1), for the calibrated model the spatial distribution of the redox interface is then assessed (section 3.1.1); 2. DAISY adopts calibrated soil parameter from MIKE SHE, additional flow and crop parameters are manually calibrated to catchment scale water balance and crop yield (section 3.2); 3. The combined DAISY-MIKE SHE nitrate model is manually calibrated using the nitrate arrival percentage (section 3.3) by adjusting the depth of the redox interface layer. The modelling results are then used to create the nitrate reduction maps (section 3.4) for a range of land use and climate scenarios (section 3.5).

## 3.1 MIKE SHE setup and calibration (Phase 1)

The calibration and setup of the MIKE SHE model has been reported in detail by Karlsson et al. (2016), but will be described here for clarity. MIKE SHE is a fully distributed hydrological model that is built as a modular system with configurable

complexity of the different flow compartments (Abbott et al., 1986; Graham and Butts, 2005). To obtain consistency with the flow calculations in DAISY, the present MIKE SHE is also based on a 1D finite difference description of the unsaturated zone using Richards' equation, parameterized by van Genuchten (1980) formulations of the retention curve and the unsaturated hydraulic conductivity curve. A 3D finite difference description of the saturated zone based on Darcy's equation was selected, including a linear reservoir formulation of tile drainage flow. The study is heavily tile drained, and it can be assumed that drainage will always be present when it is needed in the agricultural areas. However, the actual site-specific location of tile drains is unknown and therefore drains are specified across the entire catchment at a depth of -0.5 meters. Drain flow is however only activated when groundwater level rise above drain level. Apart from representing tile drainage the drainage system also represents small ditches and stream, too small to incorporate into the river system following the approach of Troldborg et al. (2010). River flow is described using the MIKE 11 module, where a relatively simple routing method (Muskingum) is used. Additionally, physical formulations for evapotranspiration (Kristensen and Jensen, 1975) and 2D overland flow (DHI, 2019) are selected. The MIKE SHE model was calibrated using the build-in autocalibration scheme AutoCal (Madsen, 2000), which uses the global search function Population Simplex Evolution method. Calibration was carried out against data from four discharge stations and 455 groundwater wells with hydraulic head measurements from the period 2004-2007 and validated in the periods 2000-2003/2008-2009. The multi-objective function consists of water balance error (mean daily error) for the four discharge stations and RMSE for the hydraulic head measurements and three of the discharge stations (one discharge station, ST45.28, is omitted here due to human regulation on the flow system). After a sensitive analysis on 28 free parameters with 43 tied parameters; a total of five parameters were chosen for calibration, of these, one soil parameter in the unsaturated zone, one drainage parameter and three saturated zone parameters.

### 3.1.1 Estimation of the redox interface

After the calibration of MIKE SHE, the estimation of the redox interface was done using a method developed by Hansen et al. (2014a). This five step method is used to determine the redox interface for each cell in MIKE SHE. . The method is based on the assumption that the present location of the redox interface is a result of the cumulative oxygen percolation through the soil column since the last ice age in the Holocene 11,700 BP.

The redox interface is assumed to have been at ground level at the end of the glaciation and to have migrated downwards by an unknown number of millimetres per yearly recharge. Following the procedure of Hansen et al. (2014a), first step is therefore to find the average yearly recharge, by running a model simulation without anthropogenic influences (abstraction and tile drainage). In the second step, the different redox capacities in soils are accounted for, where the capacity of sandy soils are multiplied with a factor of three compared to value specified for the clayey soil types applying the classification from Børgesen et al. (2013) and Greve et al. (2007). The third step generates the redox interface expressed through equation (3):

$$Redoxdepth_i = flux_i * f + min.redoxdepth$$

where $Redoxdepth_i$ is the redox depth (m) calculated at each grid (i), $flux_i$ is the groundwater recharge (m/yr.) which is multiplied by the migration constant f (yr.) (Hansen et al., 2014a). The upper part of the unsaturated zone is assumed to have no redox capacity due to very fast air diffusion, which is accounted for using a minimum redox depth, min.redoxdepth (m). To account for unrealistically high values of the redox depth, a maximum redox depth is also estimated based on the principles of Hansen et al. (2014). Hereby, the spatially distributed redox interface layer is quantified and incorporated into MIKE SHE (step four). The final step (five) is the calibration of the depth of the redox interface layer also described as the calibration of the nitrate model by adjusting the location of the layer. Step five will be described in section 3.3.

### 3.2 DAISY setup and calibration (Phase 2)

DAISY provides a one-dimensional finite difference description of soil-water-crop-atmosphere processes (Abrahamsen and Hansen, 2000; Hansen et al., 1991) where flow is described using Richards' equation and to a minor extent by macro pore flow for loamy soils (Hansen et al., 2012a). Nitrate transport is driven by the convection-dispersion algorithm in the soil matrix. The one-dimensional DAISY model can be used to represent an entire catchment by combining multiple columns of DAISY computations and evaluating the summed water balance. Calibration on catchment scale is however not straightforward and must be done manually and in an incremental manner, due to the large number of 1D model columns.

The DAISY water balance module used in this study is based on a previous calibration of the catchment (Børgesen et al., 2013), where root zone leaching and groundwater abstraction is compared with river discharge (Børgesen et al., 2013; Refsgaard et al., 2011), and subsequently extended on as described in Karlsson et al. (2016) and in the following section. The model setup for the Odense catchment contains on roughly 12,000 1D DAISY columns, and the is model is setup so that each column represents unique combinations of soil type, climate, crop rotation and groundwater depth. The DAISY model uses the same climate input and soil parameter setup as MIKE SHE and the sensitive and calibrated unsaturated soil parameter from MIKE SHE was therefore transferred to DAISY. . The water balance performance of DAISY was evaluated in the same calibration (2004-2007) and validation periods (2000-2003/2008-2009) as MIKE SHE.

In this study the DAISY model was also used to simulate nitrate leaching for each soil column that represents a unique combination of soil type, climate, crop rotation and groundwater depth. Crops are fertilized with mineral and organic nitrogen dependent on the farm type and soil type. The crop recommended nitrogen rate based on soil type and crop sequence from Danish Ministry on Agriculture (Plantedirektoratet, 2005) for the years 2004-2007 was used to setup the fertilization scheme. Nitrate leaching input are simulated on daily basis based on the leaching from the permutated crop rotations simulated for the dominating soil type within a 200m x 200 m square grid (Karlsson et al., 2016). Because of the close feedback mechanism between nitrogen yields and nitrate leaching, the simulated mean nitrogen yields were recalibrated to observed annual mean nitrogen yields on Funen (Statistikbanken, 2015) for the dominating soil type for the period 2004-2007. The calibration is conducted by adjusting the crop parameters, following the methodology of Styczen et al. (2004). Nitrogen concentrations of yields were extracted from table values of mean nitrogen contents for different crops (Møller et al., 2005). For crop rotations including clover grass and peas nitrogen biological fixation is calculated using Høgh-Jensen et al. (2004) and nitrogen atmospheric deposition is included as input to the soil using standard DAISY settings (given in Hansen et al., 2012b) for dry and wet deposition.

In the simulations under climate change, the effect of change $CO_2$ concentration in the atmosphere has an impact on the light saturated photosynthesis rate (Fm parameter), which is a crop parameter in the DAISY crop model code. In order to deal with this feedback mechanism, the procedure to change the Fm parameter was adopted from Børgesen and Olesen (2011).

### 3.3 Calibration of the nitrate model (Phase 3)

Following the approach from Hansen et al. (2014b), the nitrate model is then constructed by combining the two models, DAISY and MIKE SHE. Daily values of nitrate flux from DAISY serves as input to MIKE SHE, where nitrate transport is simulated by converting nitrate input to particles using the particle-tracking module. Each time the accumulated input of nitrate reaches 0.5 kg N within the model cell (200mx200m), a particle is released from the water table and is allowed to follow the groundwater flow. If the particle penetrates the redox interface, the nitrate is assumed to be removed completely and

instantaneously by denitrification (Hansen et al., 2014a; Postma et al., 1991). Remaining particles will emerge in discharge zones typically located in stream valleys and leave the catchment at the river outlet (Kratholm station). The nitrate arrival percentage (NAP) is found as the cumulative amount of nitrate leaving the catchment divided by the amount released at the water table.

The nitrate model is then calibrated by adjusting the depth to the redox interface through the calibration of f and min.redoxdepth to obtain the observed NAP (step five). As the calibration of these two parameters may result in non-uniqueness, all possible combinations (realisations) of the two parameters resulting in observed NAP, are identified. For all realisations the cumulative distribution of the redox depth is found at the location, where observations of redox depth are available from boreholes, as well as the cumulative distribution of the entire catchment. These two graphs are subsequently compared with the cumulative

distribution of the actual measured redox depth in boreholes. The realization with the best representation of the fractional distribution of the observed redox depth for both on-site and especially catchment scale is chosen for the final redox depth parameters.

The reason for comparing calculated redox depths to cumulative distributions for actual measurement locations and the entire catchment distribution is due to several issues. First, measured redox depths are very local point measurements, and large variations in space (within a few meters) are often reported (e.g. Ernstsen, 1996; Hansen et al., 2008), and a measurement may not be representative for the area or model scale, where numerous measurements together are more likely to represent to the correct fractional distributions in the catchment. Furthermore, the calculated redox depth may be applicable on catchment scale, on which scale it is also calibrated, but less trustworthy on location scale.

**3.4 Estimation of reduction map and map correction**

The reduction map quantifies the nitrate reduction potential for each model grid (Hansen et al., 2014a). The number of representative particles released at each cell that is subsequently reduced is divided by the total number of particles released. The reduction map is therefore based on results from the nitrate model that is run with the calibrated redox interface. In the reduction map, a grid cell with the value of 100% indicates that all particles released in the cell are subsequently reduced, while a value of 50% indicates that half of all particles (nitrate) reaches surface waters unreduced. Therefore, it provides valuable information on what the environmental impact in terms of nitrate loading of farming practices are for specific parts of the landscape.

Unfortunately, during the release of particles in the MIKE SHE model, numerical issues occasionally cause some particles to get stuck in the unsaturated zone. MIKE SHE is a commercial modelling tool and therefore there is no possibility to access the modelling code in order to correct this numerical error, or in any other way account for this model limitation. Therefore, it was necessary to introduce a correction scheme. The actual fate of these stuck particles (reduced/non-reduced) are unknown. At an early stage the assumption was made that the captured particles, if they had moved correctly through the system, would be subject to a fate similar to the non-captured particles, i.e., that the relationship between reduced/non-reduced was the same. If this assumption is valid the calculation the reduction potential in each grid cell is the same with/without the stuck particles. Unfortunately, this assumption may not always be valid. Furthermore, the arrival percentage estimated by the two methods are not the same as not all particles are released in the complex particle arrival count, the data from which it is the only way to

calibrate the nitrate model. For the two methods to be comparable it is therefore necessary to exclude the particles that are stuck in the unsaturated zone. The correction factor is therefore introduced to eliminate the particles that are stuck from changing the reduction map. The correction uses a simple linear equation, where a correction factor is manually fitted so that the arrival percentage (originating from the reduction map multiplied by the nitrate input) matches the particle arrival percentage. These corrections are done individually for all reduction maps, and the correction causes a change in the reduction in the range of -7% to 9% with a mean of 2%.

270

**3.5 Climate and land use scenarios**

One emission scenario, the IPCC AR4 SRES A1B scenario (Nakicenovic et al., 2000) where chosen as the basis for this study. Since the study was conducted newer generations of emission scenarios have been developed by the IPCC (van Vuuren et al., 2011), known as the Representative Concentration Pathways (RCPs); the A1B scenario is generally comparable to the RCP6.0-emission scenario (medium scenario).

In this study, realizations from four climate model combinations, GCM-RCM couplings, were selected from the ENSEMBLES project (Hewitt and Griggs, 2004), where results from the period 2080-2099 were extracted and used as input to the hydrological model. The reference evapotranspiration is calculated using FAO Penman– Monteith formula adapted by Allen et al. (1998) based on the climate model outputs for minimum and maximum temperature, incoming long and short wave solar radiation, relative humidity and wind speed. Following the recommendations in Allen et al. (1998) and Seaby et al. (2013) variables needed for the Penman– Monteith formula e.g., net radiation (calculated from the net incoming short and long wave radiation), water vapour pressure, height-adjusted wind speed and atmospheric pressure, where calculated from these outputs. Precipitation data from both this period and the reference period, 1990-2009, were bias-corrected (downscaled) using the DBS method, which is a direct method that preserves the dynamics and non-stationary nature of the raw climate model results (Seaby et al., 2013). While reference evapotranspiration was downscaled using a bias removal method (Seaby et al., 2013). The four selected realizations represent a wet, +19% in precipitation (ECHAM-HIRHAM5), a dry, -11% decrease in precipitation (ARPEGE—RM5.1), a warm, +3.4 °C temperature increase (HadCM3-HadRM3) and a model representing a

median projection, +10% in precipitation and +2.1 °C in temperature (ECHAM5-RCA3). The change factors can be seen in Table 1. Both climate models and bias-corrections are described in more detail in Karlsson et al. (2016).

**Table 1: Change factor for the four climate model combinations for precipitation (multiplicative), temperature (°C - additive) and reference evapotranspiration (multiplicative)**

| Season mean | Climate model | CHANGE FACTOR | | |
|---|---|---|---|---|
| | | Precipitation | Temperature | RefET |
| Annual | ARPEGE–RM5.1 | 0.88 | 2.14 | 1.12 |
| | ECHAM5–HIRHAM5 | 1.28 | 2.08 | 0.94 |
| | ECHAM5–RCA3 | 1.17 | 2.22 | 0.94 |
| | HadCM3–HadRM3 | 1.00 | 3.72 | 1.19 |
| Fall | ARPEGE–RM5.2 | 0.74 | 2.05 | 1.23 |
| | ECHAM5–HIRHAM6 | 1.20 | 2.45 | 1.02 |
| | ECHAM5–RCA4 | 1.18 | 2.35 | 0.98 |
| | HadCM3–HadRM4 | 0.93 | 4.11 | 1.33 |
| Winter | ARPEGE–RM5.3 | 1.18 | 2.40 | 1.24 |
| | ECHAM5–HIRHAM7 | 1.30 | 2.65 | 1.22 |
| | ECHAM5–RCA5 | 1.30 | 2.64 | 1.03 |
| | HadCM3–HadRM5 | 1.31 | 4.19 | 1.57 |
| Spring | ARPEGE–RM5.4 | 0.95 | 1.83 | 1.03 |
| | ECHAM5–HIRHAM8 | 1.33 | 1.72 | 0.92 |
| | ECHAM5–RCA6 | 1.18 | 2.03 | 0.88 |
| | HadCM3–HadRM6 | 1.02 | 3.24 | 1.07 |
| Summer | ARPEGE–RM5.5 | 0.67 | 2.29 | 1.14 |
| | ECHAM5–HIRHAM9 | 1.32 | 1.50 | 0.90 |
| | ECHAM5–RCA7 | 1.02 | 1.88 | 0.96 |
| | HadCM3–HadRM7 | 0.78 | 3.36 | 1.19 |

The four climate model realizations were combined with four land use scenarios, as well as the baseline (present) land use scenario. The land use scenarios were created during workshops with researcher, farming industries, environmental protection agencies and government representatives. During the workshops, participants identified possible paths of developments for the land use in Denmark considering the balance of agricultural marked value on one side and priorities in the society on the

other (e.g., environmental concerns or recreational use). From the workshop four scenarios that describe agricultural management in the period 2080-2099 was created, as: LU1: "Agriculture for nature", where the agricultural area is reduced to 40% of the land area through afforestation and increasing grass areas and fertilization rates are generally reduced (-40%); LU2: "Extensive agriculture" with a small 3% p.p. (percentage point) reduction in agricultural area resulting in 64% farmland; however, some of the intensive farm types (with high fertilization rates) are converted to less intensive farm types with less fertilization (total change of -60%); LU3: "Hightech agriculture", also with a small decrease in agricultural area of 3%, but with a productivity of crops that is assumed to increase, resulting in an insignificant change in the needed fertilizer inputs (0%); LU4: "Market driven agriculture", where forest and some extensive farm types are converted into intensive farming resulting in an agricultural area of 70%. At the same time, fertilization rates are increased to reach maximum production (+20%). More information on the land use scenarios can be found in Olesen et al. (2014) and Karlsson et al. (2016).

All 20 combinations of future climate projections (4) and land use (5) were specified as input to the hydrological model. The model was run for both future (2080-2099) and reference period (1990-2009) resulting in 40 scenarios. Additionally, the model was run with observed climate for the period 1990-2009 (5 scenarios using observed land use and the four land use scenarios). Hence, a total of 45 model simulations were analysed (Table 2). In this paper the following terminology is used:

- Observational period: Results from a hydrological model and the rootzone model DAISY run forced with observational data in the period 1990-2009. This period covers both the calibration period (2004-2007) and validation periods (2000–2003 and 2008–2009), all of which are driven by observational data.

- Reference period: This period is used to describe specifically the climate model driven hydrological results from the period 1990-2009. Future climate model runs are always compared with results from this period for the relevant climate model to ensure that climate model biases do not dominate the results.

- Future period: The future scenario refers to climate model forced runs for the period 2080-2099.

- Baseline: The term baseline refers to results from the specific model run combination (scenario number1, Table 3), where current land use scenario (LU0) is combined with the observational climate data.

| Period | Climate | Land use (LU) scenario | | | | |
|---|---|---|---|---|---|---|
| | | Baseline | Agriculture for nature | Extensive agriculture | High-tech agriculture | Market driven agriculture |
| Observatiln al/reference period, 1990-2009 | Obs. Climate | 1 | 2 | 3 | 4 | 5 |
| | ECHAM5–HIRHAM5 | 6 | 7 | 8 | 9 | 10 |
| | ECHAM5–RCA3 | 11 | 12 | 13 | 14 | 15 |
| | ARPEGE–RM5.1 | 16 | 17 | 18 | 19 | 20 |
| | HadCM3–HadRM3 | 21 | 22 | 23 | 24 | 25 |
| Future period, 2080-2099 | ECHAM5–HIRHAM5 | 26 | 27 | 28 | 29 | 30 |
| | ECHAM5–RCA3 | 31 | 32 | 33 | 34 | 35 |
| | ARPEGE–RM5.1 | 36 | 37 | 38 | 39 | 40 |
| | HadCM3–HadRM3 | 41 | 42 | 43 | 44 | 45 |

**Table 2: Land use and climate scenario matrix showing the scenario numbers. Shaded scenarios indicate the scenarios chosen for illustration in figures 3, 4, 6, 7 and 8, and the baseline scenario.**

## 4   Results

### 4.1 Model evaluation

The first phase of the calibration scheme was the MIKE SHE model calibration. A detailed presentation and evaluation of the water quantity performance of MIKE SHE, have previously been described in Karlsson et al. (2014) and Karlsson et al. (2016). MIKE SHE was reported to have good performance in the calibration and validation periods (Figure S1) with water balance error of 3% during calibration and 6/10% in the validation periods; and RMSE value of 1.5 $m^3$/s in the calibration period and 1.2/1.5 $m^3$/s during validation for the main station.


The second phase of the calibration scheme was the calibration of DAISY. Due to the complicated setup with manual simultaneous calibration of more than 12.000 1D DAISY models, the water balance performance of the DAISY model was reported to be somewhat poorer with water balance errors of 16% in the calibration period and 3- 23% in the validation periods. The water quantity performance of DAISY is also described in more detail in Karlsson et al. (2016). The results from the

subsequent manual calibration of the nitrate leaching in DAISY using the N yields for the region, can be seen in Table 3. For all the crops in the region the DAISY model is able to reproduce the observed harvested N within a margin of 0-5 kg harvested N/ha. The DAISY model is therefore able to represent the observed values of nitrate yields to a satisfactory level on catchment scale. The catchment average root zone leaching from DAISY was 40 kg NO3-N/ha/year; as there are not direct measurement of leaching, this value cannot be directly verified. But Hansen et al. (2018) have reported values in the same order of magnitude (35.1 kg NO3-N/ha/year) for a different time period (2000-2010) using the NLES leaching model (Kristensen et al., 2003; Kristensen et al., 2008), while Højberg et al. (2017); Højberg et al. (2015a) reported values of 37.5 kg NO3-N/ha/year.

**Table 3: Observed and simulated harvested N in kg N/ha for the crop types on Funen during the calibration period 2004-2007. Grain type crops are stated as harvested N in the grain, while the remaining crops are calculated as total harvested dry matter.**

| Crop | Observed harvested kg N/ha | Simulated harvested kg N/ha |
|---|---|---|
| Spring barley | 101 | 102 |
| Winter wheat | 122 | 124 |
| Winter rape | 105 | 107 |
| Silage maize | 151 | 152 |
| Grass in rotation | 258 | 259 |
| Grass permanent | 78 | 83 |
| Sugar beets | 179 | 179 |
| Grain maize | No data | - |

The third and final phases of the calibration is the calibration of the nitrate model or the calibration of the redox interface depth using the observed nitrate arrival percentage (NAP). The observed NAP of the catchment was estimated to 35-39%. The span

is a result of the choice of time period; however, a NAP of 37% was selected as the final calibration target. Several different
combinations of the migration constant (f) and the minimum redox depth (min.redoxdepth) resulted in a NAP of 37%. The
cumulative distribution of the redox depths for all combinations are compared to the cumulative distribution of the observed
redox depth (Figure S2), and the parameter combination yielding the best representation of the fractional redox depth
distribution is identified. Based on this analysis the best combination with the correct NAP was found to be f=0.01yr and
min.redoxdepth=3m. Figure 2 (left) shows the resulting depth to the redox interface for this combination. Generally, the depth
to the redox interface is shallow, with a mean depth of 4.0 meters (standard deviation of 5.6 m). This corresponds well with
the previously reported redox depth (Koch et al., 2019b). With the calibrated redox interface depth, the resulting nitrate load
in the river is 15 kg NO3-N/ha/year, this corresponds well with the observed loading of approximately 14 kg $NO_3$-N/ha/year
measured at Kratholm station.

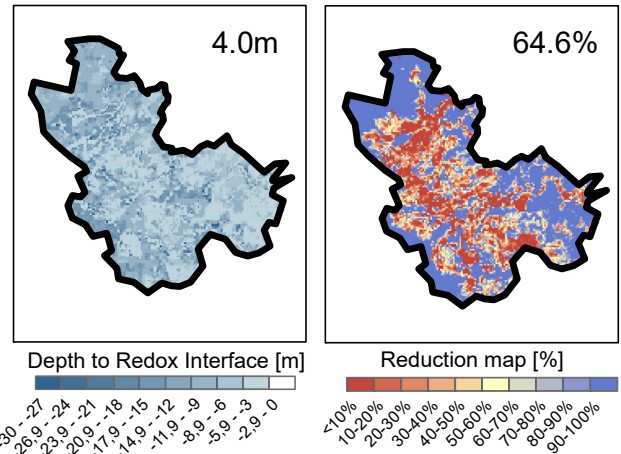

**Figure 2: Left – Resulting depth to redox interface after calibration. Right - Nitrate reduction potential maps for the baseline scenario (Land use 0/observed climate), showing the fraction of the leached nitrate that are reduced for each grid. The number in the upper right corner of each panel is the average across all grids.**

## 4.2 Baseline, reference and future nitrate reduction map

The baseline reduction map was generated based on the nitrate input from the DAISY model for the present period and the current land use (Figure 2, right). The mean nitrate reduction fraction in the catchment is 65% with a standard deviation of 38%. The spatial distribution on the map shows high reduction potentials in the uplands at the border of the model, likely where infiltrating water has a longer travel path to the river, and in areas where the redox interface is shallow. Lower reduction potential is seen in the areas near the stream network and lowland areas with deep redox interface.

Utilizing the different nitrate input from the land use/climate model scenarios, 45 nitrate models (observational, reference and future, Table 2) were run with the calibrated redox interface. The depth to the redox interface is ergo assumed to be constant in time and is not updated for each scenario. This assumption is acceptable because of the slow migration of the interface, where the migration constant predicts one meters downward movement every 100 years. Following the procedure described in section 0, this generates 45 nitrate reduction potential maps. The statistics for all 45 reduction potential maps are shown in Table 4, and a selection of these reduction maps can be seen in Figure S3 (Supplementary Material).

**Table 4: Mean and standard deviation (in brackets) across the catchment for each of the nitrate reduction potential maps (proportion of nitrate N reduced).**

| Period | Climate input | Land use | | | | |
|---|---|---|---|---|---|---|
| | | Baseline | LU1 | LU2 | LU3 | LU4 |
| Control period (1990-2009) | Obs. Climate | 0.65 (0.38) | 0.66 (0.38) | 0.65 (0.38) | 0.65 (0.38) | 0.64 (0.37) |
| | ECHAM5–HIRHAM5 | 0.62 (0.38) | 0.63 (0.38) | 0.62 (0.38) | 0.62 (0.38) | 0.62 (0.38) |
| | ECHAM5–RCA3 | 0.65 (0.37) | 0.66 (0.37) | 0.65 (0.37) | 0.65 (0.37) | 0.64 (0.37) |
| | ARPEGE–RM5.1 | 0.69 (0.36) | 0.70 (0.36) | 0.69 (0.36) | 0.68 (0.37) | 0.68 (0.36) |
| | HadCM3–HadRM3 | 0.63 (0.38) | 0.64 (0.38) | 0.61 (0.38) | 0.63 (0.38) | 0.62 (0.38) |
| Far future | ECHAM5–HIRHAM5 | 0.55 (0.38) | 0.55 (0.39) | 0.55 (0.38) | 0.55 (0.39) | 0.54 (0.38) |

| (2080-2099) | ECHAM5–RCA3 | 0.61 (0.38) | 0.62 (0.39) | 0.61 (0.38) | 0.61 (0.39) | 0.60 (0.38) |
| | ARPEGE–RM5.1 | 0.64 (0.37) | 0.64 (0.38) | 0.64 (0.37) | 0.63 (0.37) | 0.63 (0.37) |
| | HadCM3–HadRM3 | 0.67 (0.37) | 0.68 (0.37) | 0.67 (0.37) | 0.66 (0.37) | 0.65 (0.37) |

All scenarios show similar patterns of reduction zones with high and low removal fractions when compared to the baseline scenario map (Figure 2). However, although the general pattern of the reduction maps is comparable, there are spatial differences between the maps. The redox depth remains constant and all nitrate crossing this interface are assumed to be reduced, regardless of amount. Therefore, the reason for these changes in the nitrate reduction potential map results from changes in the flow path and/or changes in nitrate input from DAISY, reflected in differences between drain/interflow versus

groundwater flow to streams, and timing of nitrate release from the root zone.

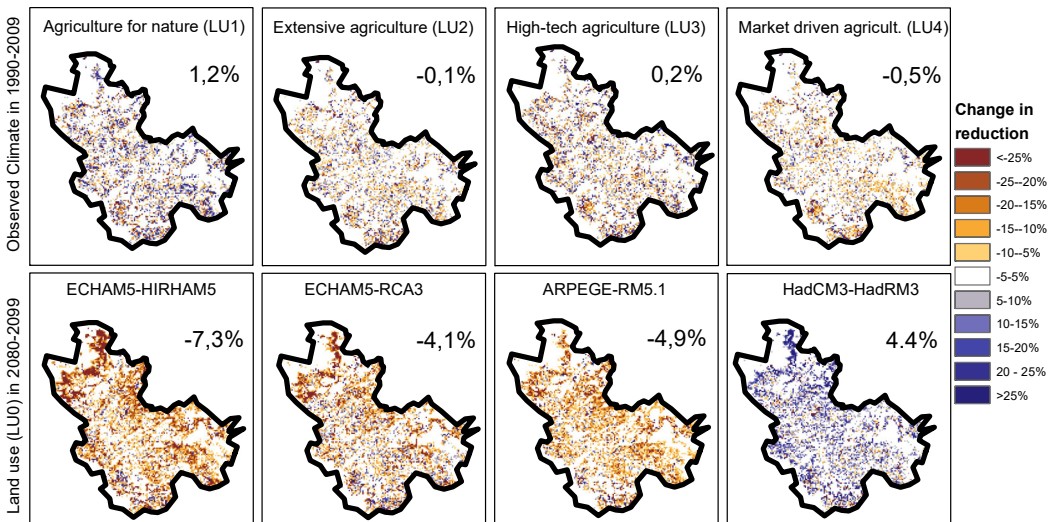

**Figure 3: Difference in nitrate reduction potential maps between: Top row - Four land use scenarios LU1-4 (Scenario 2-5) and the baseline scenario with land use 0 (Scenario 1). Bottom row: The four future climate model scenarios (scenarios 26, 31, 36, 41) and**

**the reduction maps for the corresponding climate model reference period (scenarios 6, 11, 16, 21) all for land use 0.**

**4.3 Impact of land use change on reduction maps**

To investigate the impact of land use change on the reduction maps, only land use is changed while climate remains constant, shown as the difference between land use changes scenarios and the baseline scenario (Figure 3, top row). The water balance of the models, and hence the groundwater level and water flow paths, are affected mainly by a possible change in evapotranspiration that is introduced with new crop rotation systems and vegetation types. Drains are still present in the entire catchment regardless of land use and are only active when they are required (submerged below the groundwater table). At the same time, the land use changes result in a nitrate input distribution and timing that may differ from the current land use (LU0). The changes are minor and give both higher and lower reduction potential, depending on changes in land use and vegetation for the individual grids (Figure 4, top row).

The changes in the average catchment water flow components as a result of land use change is shown in Figure 4, left. This shows the change of each component, drain flow, overland flow and base flow, when changing from a scenario run with land use 0 to the scenarios with land use 1-4. These changes are minor for the overall water balance on the catchment scale for all land use change scenarios (Figure 5, left) .

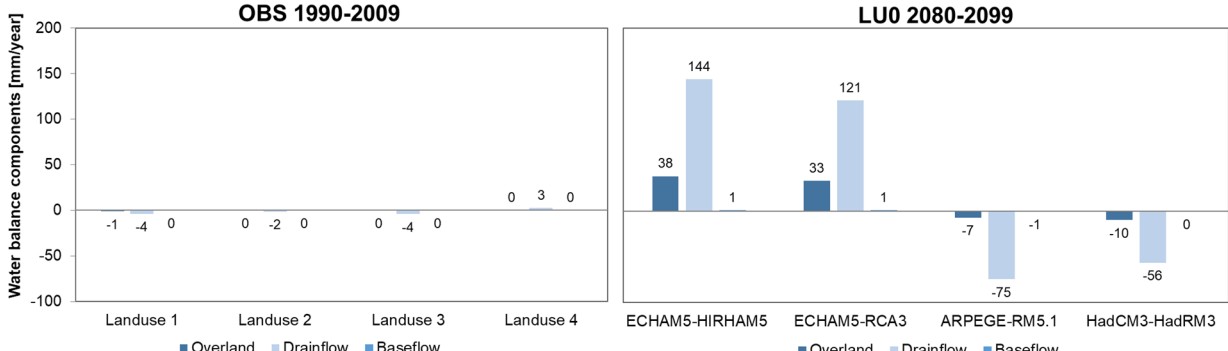

Figure 4: The change in the distribution of the water balance components in mm/year caused by changes in land use and climate change. Left – The difference for the four land use scenarios LU1-4 (Scenario 2-5) and the baseline scenario with land use 0 (scenario 1). Right: The difference between the four future climate model scenarios (scenarios 26, 31, 36, 41) and the corresponding climate model reference period (scenarios 6, 11, 16, 21) all for land use 0 (modified from Karlsson et al. (2016)).

The drain flow component is a primary conductor for non-reduced nitrate, because it represents fast and shallow flows above the redox interface. It is therefore relevant to look at the spatial distribution of the changes in drain flow for the scenarios. While the bias corrections ensure that the climate models reproduce the overall mean and variances of the observed climate, they do not necessarily ensure consistency in the temporal structure of precipitation. Hence, the overall net precipitation (precipitation-actual evapotranspiration) may change slightly across the climate models, and we have therefore plotted the change in drain flow fraction (Figure 5), defined as the drain flow divided by net precipitation, instead of the drain flow component itself. As for the changes in the reduction map (Figure 3), the differences between the reference and land use scenario drain flow fractions, are very small and sporadic. The same is found for changes in recharge and groundwater head presented in Figure S4 and Figure S5 (top row, Supplementary material). Averaged across the catchment, the change in land use gives rise to a maximum 1.2% change in reduction potential (Figure 3). For all 45 runs, the general statistics also show that changing the land use (horizontally, Table 4) does not change the mean reduction potential more than a maximum of ±2%.

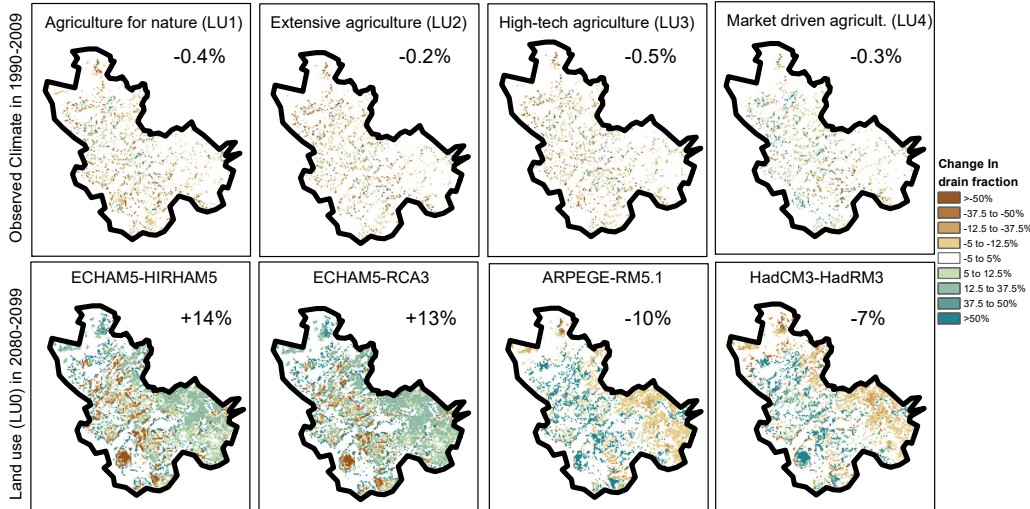

**Figure 5: Changes in the drain flow fraction from baseline (LU0) to the four land use scenarios (LU1-4) in the observational period (top row) and from the reference period to the future period for four climate models using LU0 (bottom row). Drain flow fraction is defined as drain flow divided by net precipitation. Green colors indicate that a larger percentage of the net precipitation is channelled into the drains for the future. Number in upper right corner indicate the relative change in drain flow fraction from reference period to future period. The corresponding percentage of drain flow in the baseline scenario is 31%.**

**4.4 Impact of land use and climate change on reduction maps**

Figure 3, bottom row, shows the results for the future climate compared to the corresponding reference period. For the future simulations, not only the water flow paths are affected by the change in amount and distribution of net precipitation, but also the vegetation uptake is considerably different due to increasing potential evapotranspiration. The impact of these changes is evaluated in Figure 4, right, where the changes in the flow components are shown from the reference to the future period. The climate model projections have important impacts on the distribution of the water balance components and substantial

differences are found among the models. Even if land use remains constant, the timing and amount of nitrate leakage are subject to significant variations as a result of climate changes. For the reduction maps (Figure 3), these changes result in larger differences when compared to the maps produced for the reference period. Averaged across the catchment, changes of up to 7.3% p.p. are recorded for the four scenarios and for all 45 scenarios the change is up to ±8%.

From a water balance perspective, the four future climate model projections vary greatly, but may be grouped in two overall categories. The first category includes the wet models, ECHAM5-HIRHAM5 and ECHAM5-RCA3 (Table 1). Both show a small decrease in annual reference evapotranspiration. At the same time precipitation is projected to increase in all seasons resulting in an annual increase of 15 – 30%. As a result, the net precipitation (Figure 4) and groundwater recharge (Figure S4) increase considerably. Since there is a limit to how fast water can be pushed through the deeper groundwater systems, the fast

flow components (overland, drain flow) both increase substantially.

Even though the drain flow produced with the wet climate model projections generally increases, the drain flow fraction, defined as the drain flow divided by net precipitation (Figure 5) shows two overall signals. In the eastern uphill area, the change in drain flow fractions is positive, while in the central and western areas, in the river valleys, the change in drain fraction is

negative. This implies that less net precipitation, relatively, is channelled through the drainage system than in the reference period in the river valleys (brown areas), while relatively more net precipitation is captured by drains in the eastern uphill locations. This highlights how the change in the distribution of the flow components also changes the spatial pattern across the

catchment. The underlying explanation for the pattern recognized in the wet models, is primarily found as a moderate increase in upwelling water to the drainage system in the lowlands in the future compared to the substantial increase in drainage in the uphill areas.

The increase in groundwater recharge (Figure S4) results in increasing groundwater levels (Figure S5). However, the change varies across the catchment and since flow is controlled by the gradient in hydraulic head, the non-homogeneous changes in heads will result in changes in flow direction. The general tendency is that groundwater levels increase most in the upstream parts of the catchment, while it remains the same in the valleys near the stream. Hence, the gradients will become steeper, which results in an increase in the ratio of horizontal to vertical flow. Thus, this promotes near-surface flow paths in the subsurface.

As the fast flow components, like drain and overland flow, mainly carry non-reduced water, and the subsurface reduction depends on water being transported below the redox interface for nitrate reduction to take place, both the increase in fast flow components and the shallower flow paths for the infiltrated water, is expected to give rise to a lower reduction potential as is indeed observed in Figure 3 and Table 4.

The second climate model category (dry models) includes ARPEGE-RM5.1 and HadCM3-HadRM3 (Table 1), that both show a significant increase in annual reference evapotranspiration of 10-20% (Karlsson et al., 2016). With respect to precipitation, the two models show a slight decrease or no change in annual values, whereas lower precipitation is found during summer and autumn for both models. Therefore, net precipitation decreases for both models, leading to a decrease in the fast flow components (overland, drain flow) for the future projections (Figure 4). The reduction in net precipitation also results in a reversal of the distribution of the drain flow fraction (Figure 5) and a decreasing groundwater recharge (Figure S4).

These changes are reflected in the spatial distribution of the change in groundwater level (Figure S5). As found for the wet models, the general tendency is that groundwater levels change more in the upstream parts of the catchment compared to the valleys near the stream. Therefore, the gradients will become less steep, resulting in a decrease in the ratio of horizontal to

vertical flow. This leads to a higher degree of slower and deeper groundwater flow paths, and potentially, to more nitrate crossing the redox interface.


The decrease in fast flow components and the deeper flow path supports the fact that the two models project higher reduction potential in the future. This is indeed found for one of the models, HadCM3-HadRM3, which generally has the largest increase in reduction potential of the four models, from the average reduction of 63% in the reference to 67% in the future simulation (LU0, Figure 3 and Table 4).


However, the other model, ARPEGE-RM5.1, shows a complete opposite signal with lower reduction potential (Figure 3). Initially, this anomaly did not seem to be explainable by changing flow paths as the changes (Figure 4, Figure S4 and Figure S5) are very similar to the other dry model. However, as mentioned previously, changes in the timing and quantities of the nitrate root zone leakage from DAISY is also a determining factor for the reduction map, and these dynamics are not always

apparent on average maps like the ones shown in Figure 3, Figure 4, Figure S4 and Figure S5. A closer inspection of the monthly changes in flow components and nitrate leakage provides a possible explanation for this phenomenon (calculations not shown).

The ARPEGE-RM5.1 model shows the absolute largest increase in nitrate leakage in January. Even though the model is

generally dry, the January precipitation is projected to increase in the future. The dynamics in the model shows a shift in flow components for this month towards a smaller amount of drain flow and recharge, while overland flow increases, probably as a result of larger prolonged rainfall events. This shift combined with the very large increase in nitrate leakage leads to more non-reduced water in the system for the future period than for the control period. As this combination of larger nitrate leakage and flow components shifts are not seen in the HadCM3-HadRM3 model, it is, most likely, a result of the lower reduction

potential for ARPEGE-RM5.1 in the future in spite of the overall drying signal from the model.

**4.5 Effect on the nitrate flux using different reduction maps**

To evaluate the impacts of using a fixed reduction map versus different maps calculated for each scenario explicitly considering differences caused by changes in land use and climate change, the total catchment nitrate load (nitrate arrival at Kratholm) was calculated using two different approaches for all scenarios:

1. A fixed reduction map (the baseline reduction map) is used and combined with projected nitrate leaching from the root zone.

2. Targeted reduction maps, i.e., different reduction maps for each scenario (as calculated above) are used and combined with projected nitrate leaching from the root zone.

For both approaches nitrate arrival is calculated for the observational, reference and future periods. The resulting nitrate arrivals

for approach 2 is then for each case compared to the corresponding scenario in approach 1; thereby illustrating the effect of using a targeted reduction map compared to a baseline reduction map (Figure 6).

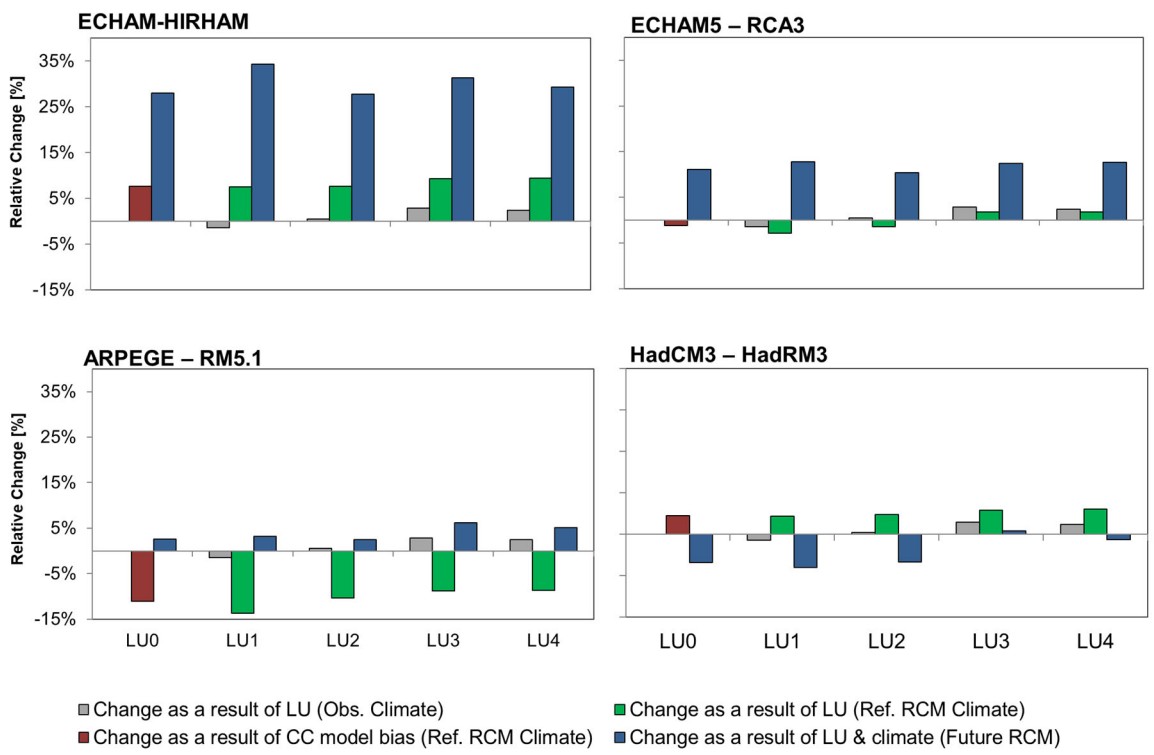

**Figure 6: Bars denote the change in nitrate flux at the catchment outlet that arises from using either a fixed nitrate reduction potential map (baseline) or using a reduction map based on the individual scenarios. The scenarios encompass cases with different land use (grey), climate model data for the present (red bar) with land use changes (green bars) or future climate data and land use changes (blue bars).**

The grey bars in Figure 6 show the relative change (compared to baseline), when applying targeted reduction maps for the four land use scenarios but using observed climate. Note therefore that the grey bars are the same for all four plots. The effect of the targeted reduction maps versus the fixed reduction map manifests in only limited spread in the estimated nitrate arrivals by only 1% - 3%. This implies that for a case of changing land use, the fixed reduction map is a reasonable approximation of the reduction potential in the catchment.

The green bars in Figure 9 represents the same effect as for the grey bars (targeted versus fixed reduction maps); however, here the climate is the reference climate for each of the four climate models. For all models, the effect is here larger than using the observed climate. This phenomenon is mainly due to the inherent bias of the reference climate model simulations. This becomes clear when looking at the first bars of each plot (red bars), denoting the change when only observed climate is replaced by reference climate but maintaining the current land use setup (LU0). Even though the climate model output is bias corrected such that the general statistics in the reference period resembles those of the observations, there may still be differences in the temporal structure of the climate model outputs, which may impact hydrological simulations. This is an important issue to bear in mind when analysing the future signal presented by the blue bars. However, the effect of land use can still be approximated by comparing the results from LU0 to LU1-4 for the different climate models. Again, the magnitude of the effect is between 1% - 3% change in nitrate arrival; however, the models do not agree on which land use causes the largest change (being either LU1 or LU4).

For the blue bars the differences from the fixed reduction map to targeted reduction maps become a combination of bias correction limitations, land use change and climate change. Even though it is not possible to completely separate the signal of

these three components, a cautious estimation can again be achieved by subtracting the blue bars with the red bar result from LU0 in the reference period, so that the signal from the climate model bias is tried to be removed. For a dry model like ARPEGE-RM5.1 the effect (land use and climate) on the nitrate arrival is in the range of -5% to -9%, while for HadCM3-HadRM3, it is between -4% to 4%. The largest effects are found for the wet models, with changes ranging from 20% - 27% for ECHAM-HIRHAM, followed by ECHAM-RCA3 with 9% - 12%. This shows that the consequences of using a fixed reduction map may be considerable, in particular with large changes in climatic conditions. Across all climate models, the average absolute effect on nitrate arrival is 10%, when using a fixed reduction map compared to a targeted reduction map.

## 5 Discussion

### 5.1 Nitrate reduction maps are not constant in time

Our analysis clearly demonstrates that nitrate reduction maps are a result of complicated interactions between climate, vegetation, geology and farm management leading to a diversity of potential nitrate inputs, distributions, timing, flow paths and reduction capability. This implies that nitrate reduction maps calculated for present climatic conditions and flow patterns will differ from nitrate reduction maps under future climate and land use conditions. The main factor causing this is differences in the precipitation/evapotranspiration regime that results in differences in how large a fraction of the water percolating from the root zone reaches the stream via shallow flow routes above the redox interface, such as overland flow and runoff via drainpipes, and how large a fraction that takes a route through deeper groundwater zones and crosses the redox interface. Therefore, nitrate reduction maps also differ between a wet year and a dry year in the present climate.

Compared to climate effects, the impacts of land use change are minor, because land use change does not affect the flow regime to the same extent as variability and change in climate. This novel finding has not been recognized in previous studies of nitrate reduction maps such as Hansen et al. (2014b), Højberg et al. (2015), Andersen et al. (2016) and Refsgaard et al. (2019). Nevertheless, it is important to note that the distribution of drains in the catchment was not changed during the land use change scenarios. Hence, uniform drain distribution and parameterization are assumed, where the drainage component

covers both natural (ditches and small canals) and agricultural drains. A different approach could have been adopted in the land use change scenarios by changing the drainage efficiency (by adjusting drainage parameters) in re- or deforested areas.

Unfortunately, little information is available to guide this fine-tuning. However, the influence of land use change on the water balance and therefore the nitrate reduction is expected to increase if this effect is accounted for.

To encapsulate the range of uncertainty and influences from climate and land use scenarios in this setup; all scenario combinations were used. However, one could speculate that not all combinations of land use and climate change scenarios

may be equally likely or plausible in the future, as decisions on land use application made by local farmers or through national regulations are made concurrently to adapt or mitigate changes in the climatic conditions.

## 5.2  Water management implications

The advantages of assuming a fixed reduction map are that any projected nitrate input, regardless of climate and land use

scenario, can be multiplied with the reduction map and thus provide a projected nitrate outflow estimate with little effort and time spent. However, as is shown in the present study, the assumption that the reduction map is constant in a changing environment is problematic. The good question then is how large errors are made by assuming a fixed reduction map and how should this be dealt with in water management practices.

The analysis indicates that assuming fixed reduction maps leads to small errors when dealing with land use change impacts but may lead to substantial errors (mean of 10% on catchment nitrate load) when climate change projections are included. Land use change impacts may however be underestimated as a result of the uniform drainage setup. 10% error on the reduction map may potentially have major impacts on practical water management. Considering for instance the baseline scenario in Table 4, where the average  nitrate reductions vary between 55% and 67% reduction, this implies that the net impact of a 100

kg N reduction in leaching from the root zone will vary between 45 kg and 33 kg (i.e. 30%). Such changes are larger than the effects of sophisticated mitigation measures (Hansen et al., 2017). Whether such errors are acceptable depends on the purpose and context in specific water management situations. Thus, using fixed reduction maps may well be justifiable for initial

screening purposes, while targeted reduction maps, explicitly calculated for specific scenarios, may be required for design of remediation measures having significant socio-economic impacts for stakeholders. The uncertainty of using a fixed reduction map for future scenarios should of course be seen in the context of the inherent uncertainties of the nitrate reduction maps (Hansen et al., 2014b).

### 5.3 Uncertainties and limitations

When using a hydrological model for simulating impacts of changes in catchment conditions compared to those existing in the calibration period, split sample validation tests are not sufficient to document a model's capability to simulate hydrological changes. Experience shows that models that are used for making predictions beyond conditions for which they are calibrated (such as land use or climate change in the present study), often suffer from model structural uncertainties (Refsgaard et al., 2012), equifinality or that parameters may not be transferrable in time (Thirel et al., 2015). In such situations the more comprehensive and data demanding differential split sample tests are recommended (KlemeŠ, 1986; Refsgaard et al., 2014). Due to lack of data such tests were beyond the scope of the present study. Instead, the model structural uncertainty for the present case was assessed using a multi-model approach with two additional hydrological models (Karlsson et al., 2016) suggesting that the signal coming from climate change was dominating over model structural uncertainty as far as hydrological change is concerned. Therefore, we argue that the inevitable uncertainties arising from model use beyond calibration conditions most likely are not so large that they affect our conclusions.

The present study was carried out for a groundwater dominated catchment characterized by till deposits, confined aquifers, and relatively shallow redox interfaces and phreatic groundwater tables. Furthermore, this catchment has a relatively uniform soil type distribution, dominated by clayey soils. We consider the conclusions to be applicable to catchments with similar hydrogeological conditions, while they cannot be used in groundwater dominated catchments characterized by alluvial plains without fast flow components such as overland flow and drainpipe/ditch flow and aerobic groundwater systems without a redox-interface. Similarly, the conclusions cannot be transferred to surface water dominated catchments, where the nitrate reduction takes place in streams and lakes, although we suspect that non-linearities may cause similar effects here.

In this study the location of redox interface was assumed to be the same for both present and future scenarios. While this assumption may be reasonable in relation to the slow natural migration of the redox interface caused by percolation of oxygen, the redox interface migration may be escalated by nitrate application on the land surface as reported by e.g. Böhlke et al. (2002) and Wriedt and Rode (2006). This issue cannot be addressed within the framework of this study, but the effect on the redox interface may be substantial especially for land use scenarios with high nitrate application.

The indication that errors can be up to 10% is based on only a single case study with one catchment, one model and a limited number of land use and climate change scenarios. While similar results may be found when applying the same approach for catchments governed by the same dominant flow processes and land use types, like the one investigated in this study, the error must be expected to be site and context specific and therefore causes projection uncertainties that should be addressed along with other known sources of uncertainty such as climate model projections, land use projections, parameter uncertainties including the effect of equifinality, geological uncertainty, and hydrological model structural uncertainty (Hansen et al., 2014b; Karlsson et al., 2016).

During calibration and validation of the model, a decrease in model performance was registered in the water balance for the validation period. This could be caused by non-optimal parameter estimates, and there is always a risk of equifinality during model calibration. In this case, the risk was minimized using an extensive dataset of both discharge, hydraulic head, redox depth and nitrate flux during calibration of the different model steps. However, a multi model set as used in this study may still be prone to the risk of equifinality. Parameter estimation are here done in a stepwise fashion for each of the models, and the catchment scale calibration of DAISY along with the particle tracking approach limits the evaluation of performance of the nitrate component to mean catchment values. The dynamic of the nitrate system is thus impossible to verify. To account for this a full solute transport solution would be necessary but was unfortunately not possible in the framework of this study; but would be a relevant next step in investigating uncertainties and improve model verification.

During simulations it was found that some particles were not correctly released due to model error, and we therefore were forced to make the assumption that the particles would be distributed similar to non-trapped particles. However, the validity of this assumption is associated with considerably uncertainty. The correction led to mean changes in reduction of 2 %, but the resulting impact on the true reduction map is unknown, as we do not know the actual travel path of the trapped particles.

## 6    Conclusions

Nitrate reduction maps are valuable tools used for calculation of remediation and climate change effects on nitrate leaching, and are generally considered constant in time, even though the timing of nitrate leaching, and flow paths may change. In this study we investigate the potential consequence for estimation nitrate climate and land use change impact projections when assuming a fixed reduction map. For an agricultural dominated catchment in Denmark, the DAISY model was used to provide nitrate leaching input, while the hydrological model MIKE SHE was used to simulate the flow regime and nitrate flow path through particle tracking. Four land use scenarios and four climate change projections were evaluated. The main finding of the study were:

- Changing climate conditions lead to reduction map changes of around 10%; whilst effects from land use changes where minor. However, land use effects may be underestimated due to drainage formulations in non-agricultural areas.

- The magnitude of the changes in the reduction map found here may, however, be influenced by both model setup (e.g., drainage), model errors (e.g., particle flow paths) and assumptions (e.g., fixed redox interface). Furthermore, the span of the chosen land use and climate change scenarios analysed and the flow regime in the study catchment may also influence results.

- The error will therefore be specific for the study site and context and it should, consequently, be tackled along with other sources of uncertainty, like geological, parameter, and model structure uncertainties that are not evaluated in this study.

**Author contributions**

IKS performed model simulations, wrote part of the paper and produced figures. TSO and JCR contributed to formulation of the conceptualization, methodology and simulation assistance, as well as the writing of the paper. DT, CDB and JEO contributed to the methodology and the writing of the paper. CDB and JEO additionally provided input data and results for the paper.

**Competing interests.**

The authors declare that they have no conflict of interest.

**Data/code availability**

The hydrological model is based on the commercial software MIKE SHE and therefore model code is not available.

Data from the study is available upon request and through the public server at https://dataverse01.geus.dk. An exception to this is the observed climate data input which is currently the property of the Danish Meteorological Institute (DMI) but will be made publicly available through https://www.dmi.dk/frie-data/ before end of 2023.

**Acknowledgements**

The present study was funded by a grant from the Danish Strategic Research Council for the Centre for Regional Change in the Earth System (CRES – www.cres-centre.dk) under contract no: DSF-EnMi 09-066868.

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
