# Peer review of "Are maps of nitrate reduction in groundwater altered by climate and land use changes?"

_Hydrology and Earth System Sciences, 2020_

## Referee Comment (RC1) · Anonymous Referee #1 · 18 Jan 2021

Summary

This study explores how different land use and climate scenarios influence nitrate reduction maps in a catchment in Denmark. Nitrate reduction maps are routinely used in Northern Europe but they are usually considered as constant in time. Because changing land use and climate will affect flowpaths and therefore the amount of nitrate crossing the redox interface, we can expect that the assumption of a constant nitrate reduction map may be wrong. This study uses a coupled model DAISY + MIKE SHE to quantify expected changes in nitrate reduction % under several scenarios. The results show that nitrate reduction maps are more sensitive to changes in the climate than

changes in land use.

General comments

This study definitely addresses an important question in the context of Northern Europe, and provides an interesting answer: catchment-scale change in nitrate reduction can reach 10% and climate changes have a greater effect compared to land use changes.

I suggest four main improvements to the manuscript:

- Place the study in a more international context. Most references are from Denmark, where the same models and maps are used. What about other countries in the Baltic area? Although I understand that nitrate reduction maps may not be common in other regions of the world, it would be interesting to do a literature review on nitrate retention and changes in flowpaths as a result of a changing climate and land use. This would greatly improve the introduction and the discussion.

- Improve the calibration strategy. This study uses a parameter-rich model, which is calibrated in one single catchment and with a poor evaluation of how the model is simulating interannual climate variability. This type of model is subject to equifinality, and it is important to evaluate it properly to make sure that it provides the right answer for the right reason. For example, would it be possible to evaluate the model in different sub-catchments with different land use % before testing land-use change scenarios? In the same way, evaluate the model in different climatic conditions during the calibration period (making the most of the past interannual climate variability) before testing future climate scenarios?

- Show time series of river discharge and nitrate concentration + model fit.

- Spatial variability and representativeness of the study area. Why only studying one catchment and not the entire country? Is it a problem of data availability, computation time? If studying more catchments is not possible, I would suggest to present the

results in different subcatchment with different land use / soil types to see if they have similar responses (preferably subcatchments with a nitrate monitoring station). This would also help assess whether the <10% change in nitrate reduction is big or not compared to current spatial variability.

Specific comments

L9: "Nitrate reduction maps have been used routinely in Northern Europe for calculating efficiency of remediation measures and impact on climate change on nitrate leaching and are as such valuable tools for policy analysis and mitigation targeting." This sentence is too long. L11&14 "Nitrate maps . . ." -> "nitrate reduction maps" L20 "The study, however, also showed that the reductions maps are products of a range of complex interactions and that the combination of the choices made for selected scenarios, model formulations and assumptions are critical for the resulting span in reduction capability." Sentence too long and unclear. Suggest to discuss whether these differences <10% are important to consider for management. How big is this 10% difference in comparison to the spatial variability across different regions of Denmark? L29 "depending on the actual hydrobiogeochemical conditions the removal may mainly occur in groundwater or in surface water systems such as lakes or wetlands" Reference needed. Please add a paragraph with references showing that the retention process studied in this paper is a dominant process in the context of the study. L75 "one of the best nutrient time series in Denmark, providing a long and near-complete data set" Please specify length, frequency. L76 "The average discharge amounts to 4.4m3/s and the load is approximately 14 kg NO3-N/ha/year" what period? Any trend? L 77 "The geology is mainly a result of previous glaciations like till deposits. Aquifers are generally confined and the phreatic groundwater tables are shallow." Move next to the sentence about soil types. L79 "There were 226 measurements of the redox depth in the area" Please briefly present the method to measured redox depth. L114 "Calibration was carried out against data from four discharge stations and 455 groundwater wells from the period 2004-115 2007, see Karlsson et al. (2016) for details" please present the

calibration strategy, objective function, parameter exploration algorithm, etc. did you calibrate Daisy and MIKE SHE together or separately? L139 "Each time the accumulated input of nitrate reach 0.5 kg N" is it 0.5 kg/ha, or kg/km$^2$? L 165 "subsequently compared with the measured redox depth in boreholes" where is this comparison? L215 "More information on the land use scenarios can be found in Olesen et al. (2014) and Karlsson et al. (2016)." Please provide a summary how these scenarios were built. l241 "For all the crops in the region the DAISY model is able to reproduce the observed harvested N" performance in evaluating the spatial variability in harvested N? l251 "To select the most appropriate combination, the cumulative distribution of the resulting redox depth of a given parameter combination is plotted against the distribution of observed redox depths. The observed redox depth is both compared to the simulated values at the actual point of observational measurement as well as to the total fractional distribution of the whole catchment (not shown). Based on this analysis the best combination with the correct NAP was found to be f=0.01yr and min.redoxdepth=3m." I found this difficult to understand, add a figure in SI? This paragraph should be moved to Materials & methods section l255 "the depth to the redox interface is fairly shallow" avoid "fairly" in a technical paper. L480 "also worth noting, that all combinations of land use and climate change scenarios may not be equally likely or even plausible in the future" Interesting remark. Please add a reference. L511"The indication that errors can be up to 10% is based on only a single case study with one catchment". Wasn't it possible to do the same study in several catchments of DK? If not, is it an issue of data availability? Computation time? If doing the same analysis in other catchments is not possible, I would suggest presenting the N retention in different subcatchments with different land use / soil type / topography.

---

## Referee Comment (RC2) · Fanny Sarrazin (Referee) · 25 Jan 2021

In the manuscript, the authors investigate the impact of changes in climate and land use on maps of nitrate reduction in groundwater in a Danish catchment. Such maps are important tools to support management strategies that deal with nitrate pollution. Therefore it is highly relevant to investigate the potential error made in current practices that use static maps and thus neglect the effect of changes in climate and land cover.

This study compares maps of nitrate reduction in groundwater produced for different climate and land use scenarios within a modelling framework. I think that a number of point need to be clarified and examined to ensure the robustness and reproducibility of

the results. I provide here a summary of my main concerns:

1) More information on the data and methods used is needed. In particular, the manuscript refer to numerous past studies for the data and methodology, which makes it difficult for the reader to have a clear understanding of the data and methods. The authors should provide in the manuscript a summary description of all data and method used (they can then refer to past studies for more details).

2) The calibration includes a number of 'manual' adjustments to the parameter values and identify a single parameterization. I think that it would be valuable to account for the uncertainty in the model parameter values and to determine whether the changes observed in the nitrate reduction maps due to changes in the climate are appreciable given the uncertainty due to parameter values. Given the presumably large number of calibrated parameters, the issue of equifinality is likely to arise, i.e. combinations of different parameter values could lead to the same model performances, but produce different nitrate reduction maps. In particular, only groundwater parameters are adjusted to match the Nitrate Arrival Percentage (NAP), and the value of these groundwater parameters could compensate for deficiencies in the values of the soil parameters (in particular soil denitrification parameters). I think this should be at least discussed in more details in the manuscript and I refer e.g. to Wade et al. (2008).

3) From the manuscript, I understand that, in the model, tile drains can be located in non-agricultural areas, which I find surprising. Some explanation on this are required, since tile drains appear to have a large impact on the model results.

I provide below detailed comments.

ABSTRACT

L20-22 'Th study, however, [...] in reduction capability: ' this sentence needs to be revised. 'complex interactions' is vague and the analyses presented in the manuscript do not explore the effect of model formulations on the nitrate reduction maps.

SECT. 1 (INTRODUCTION)

L42: I suggest citing and discussing the study by Knoll et al. (2020), that establishes a map of groundwater redox conditions for Germany through machine learning, and the study by Tesoriero (Tesoriero et al., 2015), that investigates the redox conditions in groundwater in the Chesapeake Bay watershed in the USA.

L59-60: 'A severe problem [....] resulting flow pathways.': This statement should be better explained and supported by some reference.

SECT. 2 (STUDY SITE)

L 75: Please add further details on the type, characteristics (such as frequency) of the nutrient data.

L81: A definition of the criteria to identify the redox depth (i.e. to separate aerobic from anaerobic conditions) is missing.

SECT. 3 (METHODS)

- The authors need to justify their choice of coupling the Daisy and MIKE SHE model. Why not using one model or the other? Why Daisy/MIKE SHE are particularly appropriate for this study?

- It is also not clear which parameters are calibrated and for this I think that a table that summarizes the model parameters and their calibrated value should be added (in the main text or in the supplementary information).

- What are the nitrogen inputs to the system? From L124 I understand that in cropland areas mineral fertilizers only are considered, is this correct? What about the N input for the areas with other land uses such as grass or forest? What about nitrogen biological fixation and nitrogen atmospheric deposition which can also be important inputs of nitrogen to soils?

- The grid resolution for Daisy/MIKE SHE needs to be clearly defined. Is it 200m x

200m (L 126)?

- L110-111 (and also L304-305): aren't tile drains usually located in agricultural areas? My understanding is that here tile drains are distributed uniformly independently of the land use. This is an important model assumption, since it appears that tile drainage has a large impact on the study results.

- L115 '455 groundwater wells': Which data were derived from the groundwater wells?

- L119-120 'following methods proposed by Allen et al. (1998) and Styczen et al. (2014)': it is required to add more explanation here (brief description of the methods, parameters that are calibrated with the methods). I also have the same comment regarding L128-129.

- L121 'such that they produce similar actual evapotranspiration and stream flow for the simulation period': a precise definition of what is meant by 'similar' is needed.

- L134 'the procedure described in [. . .]: Please summarize the procedure.

- L165 'compared with the measured redox depth in boreholes': a description of this comparison is missing.

- L179-181: I do not understand this statement (in particular it is not clear to me what 'direct arrival percentage' means).

- Table 1: Please define reference evapotranspiration and specify how it was calculated.

- L215-216: I would suggest to briefly summarize how these scenarios were established and to refer to Oleson et al. (2014) and Karlsson et al. (2016) for more details as currently done.

- L222 '50 model simulations': In table 3 I see only 45 and not 50 scenarios, please clarify.

[Figure]

SECT. 4 (RESULTS)

- L323: 'net precipitation' should be defined.

- L453-455: It is not clear to me how these numbers are derived from Figure 9.

SECT. 5 (DISCUSSION AND CONCLUSIONS)

- L480: replace 'the full range' by 'a range' as a limited number of scenarios were used, which cannot comprehend all possible futures.

- L515-516 'that should be addressed along with other known sources of uncertainty such as climate model projections, land use projections and hydrological model structure uncertainty.': This discussion needs to be expanded. In particular, uncertainty in model parameter values can also affect the results.

MINOR EDITS:

- L143 'section 0': please add the correct section number. - L170 and in the figure captions: add 'cell' after grid. - L230 'no 1': do the authors refer to scenario 1 in Table 3? Please clarify. - L236: replace '$m^2/s$' by '$m^3/s$'. - L345: remove 'impact' after 'change'. - L348: there is something wrong here. Maybe 'different' need to be removed? - L364-365: Please correct by 'the change in the drain flow fraction' (two occurrences). - L397 'this is also found for one of the models': remove 'also' (possibly replace by 'indeed').

REFERENCES:

Knoll, L., Breuer, L., & Bach, M. (2020). Nation-wide estimation of groundwater redox conditions and nitrate concentrations through machine learning. Environmental Research Letters, 15(6). https://doi.org/10.1088/1748-9326/ab7d5c

Tesoriero, A. J., Terziotti, S., & Abrams, D. B. (2015). Predicting Redox Conditions in Groundwater at a Regional Scale. Environmental Science and Technology, 49(16), 9657–9664. https://doi.org/10.1021/acs.est.5b01869

Wade, A. J., Jackson, B. M., & Butterfield, D. (2008). Over-parameterised, uncertain "mathematical marionettes" - How can we best use catchment water quality models? An example of an 80-year catchment-scale nutrient balance. Science of the Total Environment, 400, 52–74. https://doi.org/10.1016/j.scitotenv.2008.04.030
* * *

---

## Referee Comment (RC3) · Pia Ebeling (Referee) · 26 Jan 2021

The authors investigate how nitrate reduction is affected by climate and land use changes for one Danish catchment. This is important as nitrate reduction maps are usually considered to be constant in time, which might not be appropriate for water quality management. With a modelling approach using MIKE SHE and Daisy, the authors show that climate has a stronger impact on nitrate reduction.

General comments: - The introduction is too brief; it should inform why land use and climate changes are relevant to NO3-reduction. This is not explicitly mentioned. The key word "denitrification" also needs to be included. - The methods should be more

clearly described and structured. It is hard to follow the steps sometimes. Why are there three subsections on the nitrate model in the Method section? Is there potential to merge them? From reading the titles I do not directly know where to expect what content. Why do you define the terms in the very end of the chapter, not when talking about the data or modelling periods? Consider restructuring. - The evaluation of the model needs to be more in depth. The calibration approach needs to be better explained. Was there no calibration with nitrate concentrations? Validation results should be presented and model uncertainties in relation to the calibration/parameters need to be discussed. Maybe a sensitivity analysis would be helpful. The results are very long compared to a very short discussion and no conclusion section, consider streamlining and moving content to supplemental material. - Please revise the consistency (e.g. N, nitrate, nitrogen usage (e.g. L124) or L 126) and language (e.g. sometimes singular and plural are mixed or incomplete)

Specific comments: - L9-11 long first sentences, consider splitting. - L10: impact "of" climate change - L14: consider rewording "potential errors", what errors? This is unclear to me at this point - L20: What do the authors propose to constrain the uncertainty of model formulation and assumptions?

Introduction: - L35: "The amount of nitrate reduction occurring in groundwater depends on the flow paths and the depth to the redox interface." is very brief considering for example the Damköhler number. This part would benefit from a bit more in depth. What about availability of electron donors? - L36: "In areas with Quaternary sediments characterized by groundwater dominated flow patterns and a relatively shallow redox interface, the N-reduction in groundwater can be the dominant removal process." Please provide a reference - L43: I do not know why showing maps is considered as "a new approach". Merz et al. 2009, for example, also showed retention maps and NO3 half-life times. - L47: "produced N-reduction maps with a 100 m spatial resolution for a 101 km2 catchment in Denmark," is not helping the argumentation, can be removed. - L50: This sentence has to be checked for grammar. It is also partly redundant with

the next sentence. - L55: "the effect hence is relatively large" please be more specific, "effect" is too vague here. - L57: This sentence is not clear to me. You are saying that N-reduction maps can be used more easily than hydrological models, but actually those models are used to produce the maps. This contradicts. - L60: Please revise the sentence.

Methods: - L74: "best", "long", "near-complete" Please, specify - L76: "The average discharge amounts to 4.4m3/s and the load is approximately 14 kg NO3-N/ha/year." Reference or more details needed - L79: reformulate "There were measurements" - L84: revise "100 meter redox depth map", I assume you mean the resolution - L84: "This map" reference unclear, as citations mismatch, it is not clear to me. - L110: Please give a reference or indication why drains are needed, if locations are unknown. How do you define the "drain level" without this information? - L119 typo "percolation" - L119: "following methods" specify. Do you use several? - L121: when is the simulation period? Calibration was mentioned, but what about validation period? - L129: this is not a sentence - L135: bad title - L139: "If the particle penetrates the redox interface, the nitrate is assumed to be removed completely and instantaneously by denitrification." Please, reflect more on this assumption. - L143: section 0 - L175-181: Please, specify how you can state that this was or was not the case? How did you further investigate the stuck particles? How many particles get stuck? It seems quite a lot if the correction causes changes between -7 and 9%. - L183: Revise the sentences. Also, what was tried to improve the numerical difficulties? - L184: I did not understand how the correction was done and also why this approach was used. Please, explain. - L196: The reference seems quite old for climate projections. Thus projections for the end of this century might contain much higher uncertainty. Please explain why you used this one and not a newer study. - L197: I do not know why bias-corrections are necessary, please explain. - L200: a "combined" median model? - L210 I do not understand "3% point reduction". What is point telling here? - L218f.: I do not understand this combination that was done: "compared to the climate model results found for the reference period 1990-2009 using the same land use scenarios, resulting in 32 scenarios." I think

formulations are overcomplicated and Table 3 should be placed here.

Results: - L240: "observed trends in nitrate yields" where are these trends shown? - Table 3: I think this should be presented in the Methods? 3.6 scenarios? What does the grey shade mean? How can the climate scenarios be used for 1990-2009? Or is it necessary? I do not see them later in the presented maps. - Table 4: Do you have an idea why the standard deviations of all models are that similar (Table 4, 0.36-0.39)? Can you comment on that, please? - L295: "To investigate to what degree land use changes and climate change affect the reduction map, the difference between these scenarios and the reference scenario is shown in Figure 4." Does not seem to fit here if the next section title is "4.4 Impact of land use change on reduction maps". - Figure 3 and Figure 4 seem a bit redundant to me, considering that 4 is just the difference between the map shown in Fig2 and Fig3. I think one Figure could be economized here by merging or moving to the supplements - I think it is not necessary to show Fig7 and Fig8, especially because Fig.8 is mainly a reprint of Fig.12 in Karlsson et al. 2016, while the maps in Fig.7 do not allow to recognize more details than the general observation of land use changing little, two climate scenarios becoming wetter and two drier, which is also clear from Fig.5. Again I think redundancy should be reduced and plots removed or to the supplements.

Discussion/Conclusions: - L510: "such effects" reference unclear. Please explain further how 10% change in nitrate reduction over almost a century relates to the uncertainties of nitrate reduction maps. Is it really essential to consider changes in nitrate reduction for management, if the tool itself is already quite uncertain? - L512-516: "single case study" how representative is this case? What do you expect for other sites? Compare to other studies. General spatial differences between nitrate reduction could be considered. - L513-516: You mention uncertainties of input data (climate and land use) and model structure (though very briefly) but do not discuss the uncertainties related to the model and its parameters. - L516: This is not a nice ending. I would collect the conclusions in an extra Conclusion section.

Data/code availability: - "owned by the DMI" - what does this mean? Where and how to access it?

Figures: - F1: typo at "Market driven". Odd start of the caption as the Figure is showing the study area and land use scenarios and not a "red square". - F2: I think the map titles should be linked to the legend color bars. Maybe increase letter size of legend, add unit at left panel. To me it is unclear, what the text on the y axis refers to "Observed climate...", it seems unconnected. In the caption specify what the reduction refers to e.g. from... to.... - F5: I suggest to remove redundant subplot titles, this should be explained in the caption. - F9: The caption is unclear to me. What combinations are shown? What is meant here "or the reduction map from the scenarios"?

References: - There seems to be an error in the display of the references – is it double or some other problem?

---

## Author Comment (AC1) · 6 May 2021

Reply: Anonymous Referee #1 General comments This study definitely addresses an important question in the context of Northern Europe, and provides an interesting answer: catchment-scale change in nitrate reduction can reach 10% and climate changes have a greater effect compared to land use changes. I suggest four main improvements to the manuscript: - Place the study in a more international context. Most references are from Denmark, where the same models and maps are used. What about other countries in the Baltic area? Although I understand that nitrate reduction maps may not be common in other regions of the world, it would be interesting to

do a literature review on nitrate retention and changes in flowpaths as a result of a changing climate and land use. This would greatly improve the introduction and the discussion. Thank you for this suggestion, we have added a section in the introduction of the linkage of nitrate, Climate change and land use: It is therefore very relevant to investigate the potential error arising when nitrate reduction maps are assumed to be constant in time. No studies have been reported on that issue. Even as, the link between climate change, land use change and nitrate reduction has been established in previous studies (e.g. Fleck et al., 2017; Mas-Pla and Mencló, 2019; Olesen et al., 2019; Ortmeyer et al., 2021; Sjøeng et al., 2009). Ortmeyer et al. (2021) used a water balance model combined with a lumped-parameter nitrate mass model for an area in Germany, finding that nitrate concentrations in the groundwater increased towards the end of the century by up to 89 % as a result of changes in temperature, evapotranspiration and precipitation. Mas-Pla and Mencló (2019) found that climate change in turn affects groundwater recharge and thus the dilution of nitrate in the subsurface in a study in Catalonia. While, Paradis et al. (2016) found that new agricultural practices under changing climate conditions led to substantial nitrate increases on an Island in eastern Canada. Fleck, S. et al., 2017. Is Biomass Accumulation in Forests an Option to Prevent Climate Change Induced Increases in Nitrate Concentrations in the North German Lowland? , 8(6): 219. Mas-Pla, J., Mencló, A., 2019. Groundwater nitrate pollution and climate change: learnings from a water balance-based analysis of several aquifers in a western Mediterranean region (Catalonia). Environ Sci Pollut Res Int, 26(3): 2184-2202. DOI:10.1007/s11356-018-1859-8 Olesen, J.E. et al., 2019. Nitrate leaching losses from two Baltic Sea catchments under scenarios of changes in land use, land management and climate. Ambio, 48(11): 1252-1263. DOI:10.1007/s13280-019-01254-2 Ortmeyer, F., Mas-Pla, J., Wohnlich, S., Banning, A., 2021. Forecasting nitrate evolution in an alluvial aquifer under distinct environmental and climate change scenarios (Lower Rhine Embayment, Germany). Science of The Total Environment, 768: 144463. DOI:https://doi.org/10.1016/j.scitotenv.2020.144463 Paradis, D. et al., 2016. Groundwater nitrate concentration evolution under climate change and agricultural adaptation scenarios: Prince Edward Island, Canada. Earth Syst. Dynam., 7(1): 183-202. DOI:10.5194/esd-7-183-2016 Sjøeng, A.M.S., Kaste, Ø., Wright, R.F., 2009. Modelling future NO3 leaching from an upland headwater catchment in SW Norway using the MAGIC model: II. Simulation of future nitrate leaching given scenarios of climate change and nitrogen deposition. Hydrology Research, 40(2-3): 217-233. DOI:10.2166/nh.2009.068

- Improve the calibration strategy. This study uses a parameter-rich model, which is calibrated in one single catchment and with a poor evaluation of how the model is simulating interannual climate variability. This type of model is subject to equifinality, and it is important to evaluate it properly to make sure that it provides the right answer for the right reason. For example, would it be possible to evaluate the model in different sub-catchments with different land use % before testing land-use change scenarios? In the same way, evaluate the model in different climatic conditions during the calibration period (making the most of the past interannual climate variability) before testing future climate scenarios? Thank you for this comment. The section was unfortunately somewhat unclear, and it was not very well described the steps we took during the calibration. We have added more information on and restructured the entire the calibration procedure in the section 3.1-3.3 and inserted additional on the validation in section 4.1.

The model was calibrated in 3 steps, the parameter-rich hydrological model (Mike She), was first calibrated by identifying five sensitive parameters (among these was a single soil parameter) through a sensitivity analysis of 28 free (and 43 tied) parameters. A global search engine (Shuffled Evolution Complex) was used. Thus, limiting the risk of equifinality. Secondly, the sensitive and calibrated soil parameter was thereafter transferred to Daisy. The Daisy model has previously been manually calibrated (a manual calibration is the only way to calibrate this model due to its 1D column formulation) to the catchment, and this setup was used in this study and performance was evaluated after changing the soil parameter calibrated by Mike She. First after the calibration of these two models, the redox interface location was calibrated as a final step, using the

nitrate arrival percentage obtained at the downstream station.

It is a very valid and good suggestion to evaluate the performance of the model in different subcatchment with different land use %. The most important land use difference for nitrate is forest versus agricultural areas, but unfortunately this catchment is very agriculturally dominated, and therefore does not contain large enough forest areas to give a strong enough difference in the signal for different subcatchments. Furthermore, the only discharge station close to a larger forested area, is very doubtful as it is highly regulated and therefore measurements are uncertain. Thank you for pointing out the very relevant comment on testing the model's climate variability using dry and wet periods. This is very good idea. Unfortunately, our studies area has a generally homogeneous precipitation time series during the model simulation. Any relatively low/wet precipitation years are not occurring successively, but are isolated in the time series, making it difficult to select relatively dry/wet periods.

- Show time series of river discharge and nitrate concentration + model fit We have added a graph showing the main station hydrograph as well as the performance statistics of the calibration and validation for water balance and RMSE in the supplementary material. The way the model is setup there is no time series of nitrate concentrations given as output of the model, but a comparison of the resulting nitrate root zone leaching to other studies has been added in L338. And a comparison of the nitrate load at Kratholm to the measurements has also been added in the text (L355).

- Spatial variability and representativeness of the study area. Why only studying one catchment and not the entire country? Is it a problem of data availability, computation time? If studying more catchments is not possible, I would suggest to present the results in different subcatchment with different land use / soil types to see if they have similar responses (preferably subcatchments with a nitrate monitoring station). This would also help assess whether the <10% change in nitrate reduction is big or not compared to current spatial variability. Due to both computational time and data availability it was not possible to run the setup for more than one catchment. Especially

Daisy modelling is very time consuming as all combinations of climate/land use and soil type must be setup as an individual 1D model, which then inputs to the different grids in Mike She. Unfortunately, it is not straightforward to evaluate results from different sub-catchments. Most of the sub-catchments are very similar in relation to land use and soil type, and there are no nitrate station data available within the catchment to support the validity of reduction percentage on a subcatchment level. We have, however, added some reflection on the importance of the 10 % change in nitrate reduction compare to other studies of mitigation effect see comment for L20.

Specific comments L9: "Nitrate reduction maps have been used routinely in Northern Europe for calculating efficiency of remediation measures and impact on climate change on nitrate leaching and are as such valuable tools for policy analysis and mitigation targeting." This sentence is too long. Good point. This sentence has been split into: Nitrate reduction maps have been used routinely in Northern Europe for calculating efficiency of remediation measures and impact of climate change on nitrate leaching. These maps are therefore valuable tools for policy analysis and mitigation targeting. L11&14 "Nitrate maps . . ." -> "nitrate reduction maps" Thank you. This has been corrected. L20 "The study, however, also showed that the reductions maps are products of a range of complex interactions and that the combination of the choices made for selected scenarios, model formulations and assumptions are critical for the resulting span in reduction capability." Sentence too long and unclear. Suggest to discuss whether these differences <10% are important to consider for management. How big is this 10% difference in comparison to the spatial variability across different regions of Denmark? Yes, this is not a clear sentence. We have changed it according to the suggestion and added reflections on the importance for management in L19: The study, however, also showed that the reductions maps are products of a range of complex interactions between water fluxes, nitrate use and timing. What is also important to note, is that the choices made for future scenarios, model setup and assumptions may affect the resulting span in reduction capability. To account for this uncertainty multiple approaches, assumptions and models could be applied for

the same area, however as these models are very time consuming this is not always a feasible approach in practice. An uncertainty in the order of 10% on the reduction map may have major impacts on practical water management. L593: 10% error on the reduction map may potentially have major impacts on practical water management. Considering for instance the baseline scenario in Table 4, where the average N-reductions vary between 55% and 67% reduction, this implies that the net impact of a 100 kg N reduction in leaching from the root zone will vary between 45 kg and 33 kg (i.e. 30%). Such changes are larger than the effects of sophisticated mitigation measures (Hansen et al., 2017). Hansen AL, Refsgaard JC, Olesen JE, Børgesen CD (2017) Potential benefits of a spatially targeted regulation based on detailed N-reduction maps to decrease N-load from agriculture in a small groundwater dominated catchment. Science of the Total Environment, 595, 325-336. L29 "depending on the actual hydrobiogeochemical conditions the removal may mainly occur in groundwater or in surface water systems such as lakes or wetlands" Reference needed. Please add a paragraph with references showing that the retention process studied in this paper is a dominant process in the context of the study. Thank you for this comment, we have added a reference to sentence mentioned above: It can be expressed as a percentage removal and depending on the actual hydrobiogeochemical conditions the removal may mainly occur in groundwater or in surface water systems such as lakes or wetlands (Huno et al., 2018; Quick et al., 2019). Huno, S.K.M., Rene, E.R., van Hullebusch, E.D., Annachhatre, A.P., 2018. Nitrate removal from groundwater: a review of natural and engineered processes. Journal of Water Supply: Research and Technology-Aqua, 67(8): 885-902. DOI:10.2166/aqua.2018.194 Quick, A.M. et al., 2019. Nitrous oxide from streams and rivers: A review of primary biogeochemical pathways and environmental variables. Earth-Science Reviews, 191: 224-262. DOI:https://doi.org/10.1016/j.earscirev.2019.02.021

And a reference in L84, about the dominant process in the study area: The reduction in this catchment has previously been shown to be dominated by saturated zone reduction processes (Hansen et al., 2009). Hansen, J.R. et al., 2009. An integrated

and physically based nitrogen cycle catchment model. Hydrology Research, 40(4): 347-363.

L75 "one of the best nutrient time series in Denmark, providing a long and near-complete data set" Please specify length, frequency. More information on the station has been added: The discharge station at Kratholm has one of the best nutrient time series in Denmark starting in the 1980s, with near-daily sampling from 1989 (Windolf et al., 2016). The station, therefore, provides a long and near-complete data set for nutrient modelling as well as an extensive water discharge time series (Trolle et al., 2019). Trolle, D. et al., 2019. Effects of changes in land use and climate on aquatic ecosystems: Coupling of models and decomposition of uncertainties. Science of The Total Environment, 657: 627-633. DOI:https://doi.org/10.1016/j.scitotenv.2018.12.055 Windolf, J. et al., 2016. Successful reduction of diffuse nitrogen emissions at catchment scale: example from the pilot River Odense, Denmark. Water science and technology : a journal of the International Association on Water Pollution Research, 73(11): 2583-9. DOI:10.2166/wst.2016.067

L76 "The average discharge amounts to 4.4m3/s and the load is approximately 14 kg NO3-N/ha/year" what period? Any trend? More information has been added here: In 2005-2009, the average discharge amounts to 4.6m3/s and the load is approximately 14 kg NO3-N/ha/year. A decreasing trend in nitrate loads has been observed previously during 2000-2013 by Windolf et al. (2016), possibly due to implementation of mitigation measures in the catchment. Windolf, J. et al., 2016. Successful reduction of diffuse nitrogen emissions at catchment scale: example from the pilot River Odense, Denmark. Water science and technology : a journal of the International Association on Water Pollution Research, 73(11): 2583-9. DOI:10.2166/wst.2016.067

L 77 "The geology is mainly a result of previous glaciations like till deposits. Aquifers are generally confined and the phreatic groundwater tables are shallow." Move next to the sentence about soil types. The sentence has been moved. L79 "There were 226 measurements of the redox depth in the area" Please briefly present the method to

measured redox depth. Thank you, a sentence has been added on this: 226 measurements of the redox depth are available from boreholes in the area, the redox depths were mainly interpreted based on sediment color as described by e.g. Ernstsen and Mørup (1992), and a few by measurements of reduced compounds. Ernstsen, V., Mørup, S., 1992. Nitrate reduction in clayey till by Fe(II) in clay minerals. Hyperfine Interactions, 70(1): 1001-1004. DOI:10.1007/BF02397497

L114 "Calibration was carried out against data from four discharge stations and 455 groundwater wells from the period 2004-115 2007, see Karlsson et al. (2016) for details" please present the calibration strategy, objective function, parameter exploration algorithm, etc. did you calibrate Daisy and MIKE SHE together or separately? Thank you for this comment. The method section is indeed somewhat confusing. We have tried to structure it more clearly by moving section and paragraphs and adding some introductory sentences (L121-L137). We have also added more information in the entire section. L139 "Each time the accumulated input of nitrate reach 0.5 kg N" is it 0.5 kg/ha, or kg/km2? This has been clarified in the paper: Each time the accumulated input of nitrate reach 0.5 kg N within the model cell (200mx200m), a particle is released from the water table and is allowed to follow the groundwater flow. L 165 "subsequently compared with the measured redox depth in boreholes" where is this comparison? This line has been deleted. See reply for L251. L215 "More information on the land use scenarios can be found in Olesen et al. (2014) and Karlsson et al. (2016)." Please provide a summary how these scenarios were built. Thank you for pointing out this missing part. We have added more information in line 289-293: The land use scenarios were created during workshops with researcher, farming industries, environmental protection agencies and government representatives. During the workshops, participants identified possible paths of developments for the land use in Denmark considering the balance of agricultural marked value on one side and priorities in the society on the other (e.g., environmental concerns or recreational use). From the workshop four scenarios that describe agricultural management in the period 2080-2099 was created, l241 "For all the crops in the region the DAISY

model is able to reproduce the observed harvested N" performance in evaluating the spatial variability in harvested N? Thank you for this comment. Unfortunately, there a no information available to evaluate the harvested N spatially, as information on this is only registered on a regional level. Therefore, we cannot test the Daisy performance on the spatial variability. l251 "To select the most appropriate combination, the cumulative distribution of the resulting redox depth of a given parameter combination is plotted against the distribution of observed redox depths. The observed redox depth is both compared to the simulated values at the actual point of observational measurement as well as to the total fractional distribution of the whole catchment (not shown). Based on this analysis the best combination with the correct NAP was found to be f=0.01yr and min.redoxdepth=3m." I found this difficult to understand, add a figure in SI? This paragraph should be moved to Materials & methods section Thank you, yes this is unclear. The paragraph above has been moved to section 3.4. We have also added some more explanation (L226-239): As the calibration of these two parameters may result in non-uniqueness, all possible combinations (realisations) of the two parameters resulting in observed NAP, are identified. For all realisations the cumulative distribution of the redox depth is found at the location, where observations of redox depth are available from boreholes, as well as the cumulative distribution of the entire catchment. These two graphs are subsequently compared with the cumulative distribution of the actual measured redox depth in boreholes. The realization with the best representation of the fractional distribution of the observed redox depth for both on-site and especially catchment scale is chosen for the final redox depth parameters. The reason for comparing calculated redox depths to cumulative distributions for actual measurement locations and the entire catchment distribution is due to several issues. First, measured redox depths are very local point measurements, and large variations in space (within a few meters) are often reported (e.g. Ernstsen, 1996; Hansen et al., 2008), and a measurement may not be representable for the area or model scale, where numerous measurements together are more likely to represent to the correct fractional distributions in the catchment.

Furthermore, the calculated redox depth may be applicable on catchment scale, on which scale it is also calibrated, but less trustworthy on location scale. Ernstsen, V., 1996. Reduction of Nitrate By Fe2+ in Clay Minerals. Clays and Clay Minerals, 44(5): 599-608. DOI:10.1346/CCMN.1996.0440503 Hansen, J.R., Ernstsen, V., Refsgaard, J.C., Hansen, S., 2008. Field scale heterogeneity of redox conditions in till-upscaling to a catchment nitrate model. Hydrogeology Journal, 16(7): 1251-1266. DOI:10.1007/s10040-008-0330-1 The remaining paragraph in L251 is moved as suggested. l255 "the depth to the redox interface is fairly shallow" avoid "fairly" in a technical paper. Thank you, this has been corrected. L480 "also worth noting, that all combinations of land use and climate change scenarios may not be equally likely or even plausible in the future" Interesting remark. Please add a reference. This is more a reflection on the fact that as climate changes in one way or the other there may be some political and financial strategies that are more likely to occur than others. We have no reference for this, and have therefore modified the sentence, so that it now states: However, one could speculate that not all combinations of land use and climate change scenarios may be equally likely or plausible in the future, as decisions on land use application made by local farmers or through national regulations are made concurrently to adapt or mitigate changes in the climatic conditions. L511"The indication that errors can be up to 10% is based on only a single case study with one catchment". Wasn't it possible to do the same study in several catchments of DK? If not, is it an issue of data availability? Computation time? If doing the same analysis in other catchments is not possible, I would suggest presenting the N retention in different subcatchments with different land use / soil type / topography See response to main comment 4.

Please also note the supplement to this comment:
https://hess.copernicus.org/preprints/hess-2020-570/hess-2020-570-AC1-supplement.pdf

**Supplement:**

**Supplementary material**

[Figure]

**Figure S1: Left - Hydrograph of the main discharge station for observed and simulated flow at the outlet (Kratholm) during the calibration period. Right – Table of performance during the calibration (2004-2007), the validation period 1 (2000-2003) and validation period 2 (2008-2009). Figure modified from Karlsson et al. (2016).**

[Figure]

**Figure S2: Nitrate reduction potential maps for four land use scenarios (LU1-4) in the observational period (top row) and four climate scenarios (LU0) for the future climate (bottom row), showing the fraction of the added nitrate that is reduced for each grid. In the top row observed climate is used, changing only the land use (scenarios 2-5). In the bottom row the present land use (LU0) is used, while using future climate projected by the four climate models (scenarios 26, 31, 36 and 41).**

[Figure]

**Figure S3: Changes in daily recharge from baseline (LU0) to the four land use scenarios (LU1-4) in the observational period (top row) and the change for four climate scenarios (LU0) from reference to future period (bottom row).**

[Figure]

**Figure S4: Changes in groundwater level from the upper saturated layer. Changes are reported as change from (LU0) to LU1-4 with observational climate for the top row and from RCM reference period to RCM future period for the bottom row. Figure modified from Karlsson et al. (2016).**

---

## Author Comment (AC2) · 6 May 2021

Reply: Referee #2 General comments In the manuscript, the authors investigate the impact of changes in climate and land use on maps of nitrate reduction in groundwater in a Danish catchment. Such maps are important tools to support management strategies that deal with nitrate pollution. Therefore it is highly relevant to investigate the potential error made in current practices that use static maps and thus neglect the effect of changes in climate and land cover. This study compares maps of nitrate reduction in groundwater produced for different climate and land use scenarios within a modelling framework. I think that a number of point need to be clarified and examined

to ensure the robustness and reproducibility of the results. I provide here a summary of my main concerns:

1) More information on the data and methods used is needed. In particular, the manuscript refer to numerous past studies for the data and methodology, which makes it difficult for the reader to have a clear understanding of the data and methods. The authors should provide in the manuscript a summary description of all data and method used (they can then refer to past studies for more details). Thank you for this comment. We have added new information on the methodology and restructured the entire method section. We have also added more information on the data and the data sources in the data section (2. Study site).

2) The calibration includes a number of 'manual' adjustments to the parameter values and identify a single parameterization. I think that it would be valuable to account for the uncertainty in the model parameter values and to determine whether the changes observed in the nitrate reduction maps due to changes in the climate are appreciable given the uncertainty due to parameter values. Given the presumably large number of calibrated parameters, the issue of equifinality is likely to arise, i.e. combinations of different parameter values could lead to the same model performances, but produce different nitrate reduction maps. In particular, only groundwater parameters are adjusted to match the Nitrate Arrival Percentage (NAP), and the value of these groundwater parameters could compensate for deficiencies in the values of the soil parameters (in particular soil denitrification parameters). I think this should be at least discussed in more details in the manuscript and I refer e.g. to Wade et al. (2008). Thank you for this comment. Yes, the section was unfortunately somewhat unclear and it was not very well described the steps we took during the calibration to ensure equifinality. We have added some extra introductory sentences to the method section (L121-L137), as well as restructured the entire section and changed the headlines. We have also added more information on the calibration procedure in the section 3.1-3.3 and on the validation in section 4.1, and an additional figure (Figure S1) in the supplementary material

showing the stream hydrograph and performance.

The model was calibrated in 3 steps, the parameter-rich hydrological model (Mike She), was first calibrated by identifying five sensitive parameters (among these was a single soil parameter) through a sensitivity analysis of 28 free (and 43 tied) parameters. A global search engine (Shuffled Evolution Complex) was used. Thus, limiting the risk of equifinality. Secondly, the sensitive and calibrated soil parameter was thereafter transferred to Daisy. The Daisy model has previously been manually calibrated (a manual calibration is the only way to calibrate this model due to its 1D column formulation) to the catchment, and this setup was used in this study and performance was evaluated after changing the soil parameter calibrated by Mike She. First after the calibration of these two models, the redox interface location was calibrated as a final step, using the nitrate arrival percentage obtained at the downstream station.

3) From the manuscript, I understand that, in the model, tile drains can be located in non-agricultural areas, which I find surprising. Some explanation on this are required, since tile drains appear to have a large impact on the model results. Yes, this true. We did not explain this. We have added text on this issue in L145-149: The study is heavily tile drained, and it can be assumed that drainage will always be present when it is need in the agricultural areas. However, the actual site-specific location of tile drains are unknown and therefore drains are specified across the entire catchment at a depth of -0.5 meters . Drain flow is however only activated when groundwater level rise above drain level. Apart from representing tile drainage the drainage system also represents small ditches and stream, too small to incorporate into the river system following the approach of Troldborg et al. (2010).

Troldborg, L. et al., 2010. DK-model2009 - Modelopstilling og kalibrering for Fyn. GEUS Report.

I provide below detailed comments. ABSTRACT L20-22 'Th study, however, [...] in reduction capability: ' this sentence needs to be revised. 'complex interactions' is vague

and the analyses presented in the manuscript do not explore the effect of model formulations on the nitrate reduction maps. We have reformulated the sentence: The study, however, also showed that the reductions maps are products of a range of complex interactions between water fluxes, nitrate use and timing. What is also important to note, is that the choices made for future scenarios, model setup and assumptions may affect the resulting span in reduction capability.

SECT. 1 (INTRODUCTION) L42: I suggest citing and discussing the study by Knoll et al. (2020), that establishes a map of groundwater redox conditions for Germany through machine learning, and the study by Tesoriero (Tesoriero et al., 2015), that investigates the redox conditions in groundwater in the Chesapeake Bay watershed in the USA. Thank you, we have added these references. L49-54: Heterogeneities in geology and drainage systems are responsible for substantial local spatial variations in nitrate reduction. However, the spatial variation of nitrate reduction in the groundwater system has so far only been investigated in a handful of studies (e.g. Højberg et al., 2015a; Knoll et al., 2020; Kunkel et al., 2008; Merz et al., 2009; Tesoriero et al., 2015; Wriedt and Rode, 2006). Different approaches have been used in these studies from nitrate groundwater modelling (Højberg et al., 2015b; Merz et al., 2009; Wriedt and Rode, 2006), data driven machine learning (Knoll et al., 2020) or statistical modelling (Tesoriero et al., 2015).

L59-60: 'A severe problem [. . ..] resulting flow pathways.': This statement should be better explained and supported by some reference. We have reformulated this sentence and added a reference: A severe problem in this respect is, however, that the nitrate reduction maps may not be constant in time as the reduction taking place at a given location depend on resulting flow pathways (Hansen et al., 2014b).

SECT. 2 (STUDY SITE) L 75: Please add further details on the type, characteristics (such as frequency) of the nutrient data. Thank you, this has been added in L98-L104: The discharge station at Kratholm has one of the best nutrient time series in Denmark starting in the 1980s, with near-daily sampling from 1989 (Windolf et al., 2016). The

station, therefore, provides a long and near-complete data set for nutrient modelling as well as an extensive water discharge time series (Trolle et al., 2019). In 2005-2009, the average discharge amounts to 4.6m3/s and the load is approximately 14 kg NO3-N/ha/year. A decreasing trend in nitrate loads has been observed previously during 2000-2013 by Windolf et al. (2016), possibly due to implementation of mitigation measures in the catchment.

Trolle, D. et al., 2019. Effects of changes in land use and climate on aquatic ecosystems: Coupling of models and decomposition of uncertainties. Science of The Total Environment, 657: 627-633. DOI:https://doi.org/10.1016/j.scitotenv.2018.12.055 Windolf, J. et al., 2016. Successful reduction of diffuse nitrogen emissions at catchment scale: example from the pilot River Odense, Denmark. Water science and technology : a journal of the International Association on Water Pollution Research, 73(11): 2583-9. DOI:10.2166/wst.2016.067

L81: A definition of the criteria to identify the redox depth (i.e. to separate aerobic from anaerobic conditions) is missing. This information has been added, L105-106: 226 measurements of the redox depth are available from boreholes in the area, the redox depths were mainly interpreted based on sediment colour as described by e.g. Ernstsen and Mørup (1992), and a few by measurements of reduced compounds.

Ernstsen, V., Mørup, S., 1992. Nitrate reduction in clayey till by Fe(II) in clay minerals. Hyperfine Interactions, 70(1): 1001-1004. DOI:10.1007/BF02397497

SECT. 3 (METHODS) - The authors need to justify their choice of coupling the Daisy and MIKE SHE model. Why not using one model or the other? Why Daisy/MIKE SHE are particularly appropriate for this study? Thank you for this comment. We have added some reflections on this in L 124-130: Both Mike She and Daisy have been used extensively in the danish area, and Mike She forms the basis of the national nitrate and groundwater model (Bruun et al., 2003; Hoang et al., 2010; Højberg et al., 2015a; Højberg et al., 2010; Højberg et al., 2013; Højberg et al., 2015b; Troldborg et al.,

2010). Mike She is a fully coupled integrated groundwater-surface water model and this integration is important for assessing the feedback between unsaturated and saturated zone, especially under changing climate. However, Mike She does not simulate crops development and nitrate leaching from the root zone, and therefore information from an agrological model, like Daisy, is necessary.

Bruun, S., Christensen, B.T., Hansen, E.M., Magid, J., Jensen, L.S., 2003. Calibration and validation of the soil organic matter dynamics of the Daisy model with data from the Askov long-term experiments. Soil Biology and Biochemistry, 35(1): 67-76. DOI:http://dx.doi.org/10.1016/S0038-0717(02)00237-7 Hoang, L. et al., 2010. Comparison of the SWAT model versus DAISY-MIKE SHE model for simulating the flow and nitrogen processes. In: conference, T.I.S. (Ed.). Højberg, A.L. et al., 2015a. En ny kvælstofmodel. Oplandsmodel til belastning og virkemidler. Metode rapport (A new nitrogen model. Catchment model for loads and measures. Methodology Report – In Danish). DOI:Available from http://www.geus.dk/DK/water-soil/water-cycle/Documents/national_kvaelstofmodel_metoderapport.pdf Højberg, A.L. et al., 2010. DK-model2009 – Sammenfatning af opdateringen 2005-2009, Geological Survey of Denmark and Greenland. Højberg, A.L., Troldborg, L., Stisen, S., Christensen, B.B.S., Henriksen, H.J., 2013. Stakeholder driven update and improvement of a national water resources model. Environmental Modelling & Software, 40(0): 202-213. DOI:http://dx.doi.org/10.1016/j.envsoft.2012.09.010 Højberg, A.L. et al., 2015b. National kvælstofmodel - Oplandsmodel til belastning og virkemidler. Metode rapport, revideret udgave september 2015 (National nitrogenmodel - Catchment model for load and measures. Method report, revised version September 2015). DOI:ISBN 978-87-7871-418-3 Troldborg, L. et al., 2010. DK-model2009 - Modelopstilling og kalibrering for Fyn. GEUS Report .

- It is also not clear which parameters are calibrated and for this I think that a table that summarizes the model parameters and their calibrated value should be added (in the main text or in the supplementary information). Thank you for this comment.

The calibration and parameters have already been reported in detail elsewhere and we would therefore like to avoid going into detail with the parameters, so the the paper do not get too long. We, however, do acknowledge that some information is needed here. We have therefore added a general statement of the parameter types included in the calibration at L157: After a sensitive analysis on 28 free parameters with 43 tied parameters; a total of five parameters were chosen for calibration, of these, one soil parameter in the unsaturated zone, one drainage parameter and three saturated zone parameters.

- What are the nitrogen inputs to the system? From L124 I understand that in cropland areas mineral fertilizers only are considered, is this correct? What about the N input for the areas with other land uses such as grass or forest? What about nitrogen biological fixation and nitrogen atmospheric deposition which can also be important inputs of nitrogen to soils? Thank you for the comment, yes that was indeed not clear. We have expanded this explanation in L199-208: Daisy also simulates nitrate leaching for each soil column that represents a unique combination of soil type, climate, crop rotation and groundwater depth. Crops are fertilized with mineral and organic nitrogen dependent on the farm type and soiltype. The crop recommended nitrogen rate for the years 2004-2007 was used to setup the fertilization scheme. Nitrate leaching input are simulated on daily basis based on the leaching from the permutated crop rotations simulated for the dominating soil type within a 200m x 200 m square grid (Karlsson et al., 2016). Because of the close feedback mechanism between nitrogen yields and nitrate leaching, the simulated mean nitrogen yields were recalibrated to observed annual mean nitrogen yields on Funen (Statistikbanken, 2015) for the dominating soil type for the period 2004-2007. The calibration is conducted by adjusting the crop parameters, following the methodology of Styczen et al. (2004). Nitrogen concentrations of yields were extracted from table values of mean nitrogen contents for different crops (Møller et al., 2005). For crop rotations including clover grass and peas nitrogen biological fixation is calculated using Høgh-Jensen et al. (2004) and nitrogen atmospheric deposition is included as input to the soil using standard Daisy settings for dry

and wet deposition (Hansen et al., 2012). Hansen, S., Abrahamsen, P., T. Petersen, C., Styczen, M., 2012. Daisy: Model Use, Calibration, and Validation. Transactions of the ASABE, 55(4): 1317. DOI:https://doi.org/10.13031/2013.42244 Høgh-Jensen, H., Loges, R., Jørgensen, F.V., Vinther, F.P., Jensen, E.S., 2004. An empirical model for quantification of symbiotic nitrogen fixation in grass-clover mixtures. Agricultural Systems, 82(2): 181-194. DOI:https://doi.org/10.1016/j.agsy.2003.12.003 Karlsson, I.B. et al., 2016. Combined effects of climate models, hydrological model structures and land use scenarios on hydrological impacts of climate change. Journal of Hydrology, 535: 301-317. DOI:http://dx.doi.org/10.1016/j.jhydrol.2016.01.069 Møller, J. et al., 2005. Fodermiddeltabel - Sammensætning og foderværdi af fodermidler til kvæg. 64. Statistikbanken, 2015. Statistical regional registrated annual mean yields. (In Danish) https://www.statistikbanken.dk/jord3. Styczen, M. et al., 2004. Standardopstillinger til Daisy-modellen. Vejledning og baggrund, Institut for Vand og Miljø, DHI.

- The grid resolution for Daisy/MIKE SHE needs to be clearly defined. Is it 200m x200m (L 126)? See comment above, we have added more information on this in the cited section (L203).

- L110-111 (and also L304-305): aren't tile drains usually located in agricultural areas? My understanding is that here tile drains are distributed uniformly independently of the land use. This is an important model assumption, since it appears that tile drainage has a large impact on the study results. See response to main comment 3)

- L115 '455 groundwater wells': Which data were derived from the groundwater wells? Thanks, we added L153: and 455 groundwater wells with hydraulic head measurements from the period 2004-2007

[revised manuscript text omitted]

- L121 'such that they produce similar actual evapotranspiration and stream flow for the simulation period': a precise definition of what is meant by 'similar' is needed. See comment to L119-120

- L134 'the procedure described in [. . .]: Please summarize the procedure. See comment to L119-120

- L165 'compared with the measured redox depth in boreholes': a description of this comparison is missing. Evt. Supmat. Thank you, yes this is unclear. We have added some more explanation (L226-239): As the calibration of these two parameters may result in non-uniqueness, all possible combinations (realisations) of the two parameters resulting in observed NAP, are identified. For all realisations the cumulative distribution of the redox depth is found at the location, where observations of redox depth are available from boreholes, as well as the cumulative distribution of the entire catchment. These two graphs are subsequently compared with the cumulative distribution of the actual measured redox depth in boreholes. The realization with the best representation of the fractional distribution of the observed redox depth for both on-site and especially catchment scale is chosen for the final redox depth parameters. The reason for comparing calculated redox depths to cumulative distributions for actual measurement locations and the entire catchment distribution is due to several issues. First, measured redox depths are very local point measurements, and large variations in space (within a few meters) are often reported (e.g. Ernstsen, 1996; Hansen et al., 2008), and a measurement may not be representable for the area or model scale, where numerous measurements together are more likely to represent to the correct fractional distributions in the catchment. Furthermore, the calculated redox depth may be applicable on catchment scale, on which scale it is also calibrated, but less trustworthy on location scale. Ernstsen, V., 1996. Reduction of Nitrate By Fe2+ in Clay Minerals. Clays and Clay Minerals, 44(5): 599-608. DOI:10.1346/CCMN.1996.0440503 Hansen, J.R., Ernstsen, V., Refsgaard, J.C., Hansen, S., 2008. Field scale heterogeneity of redox conditions in till-upscaling to a catchment nitrate model. Hydrogeology Journal, 16(7): 1251-1266. DOI:10.1007/s10040-008-0330-1 - L179-181: I do not understand this statement (in particular it is not clear to me what 'direct arrival percentage' means). We have reformulated this sentence to make it more clear: If this assumption is valid the calculation the reduction potential in each grid cell is the same with/without the stuck particles. Unfortunately, this assumption may not always be valid. Furthermore, the arrival percentage estimated by the two methods are not the same as not all particles are released in the complex particle arrival count, the data from which is the only way to calibrate the nitrate model. For the two methods to be comparable it is therefore necessary to exclude the particles that are stuck in the unsaturated zone.

- Table 1: Please define reference evapotranspiration and specify how it was calculated. The reference evapotranspiration (also sometimes denoted as potential evapotranspiration) as referred to in Table 1, is calculated based on variables from the climate models. We have added information on how it was calculated in L276-277: The reference evapotranspiration is calculated using FAO Penman– Monteith formula based on the climate model outputs for temperature, radiation, water vapour wind speed and water pressure.

- L215-216: I would suggest to briefly summarize how these scenarios were established and to refer to Oleson et al. (2014) and Karlsson et al. (2016) for more details as currently done. Thank you. This has been added in L289-293: The land use scenarios were created during workshops with researcher, farming industries, environmental protection agencies and government representatives. During the workshops, participants identified possible paths of developments for the land use in Denmark considering the balance of agricultural marked value on one side and priorities in the society on the other (e.g., environmental concerns or recreational use). From the workshop four scenarios that describe agricultural management in the period 2080-2099 was created,

- L222 '50 model simulations': In table 3 I see only 45 and not 50 scenarios, please clarify. Yes this is a mistake, it has been corrected to 45.

SECT. 4 (RESULTS) - L323: 'net precipitation' should be defined. This has been added in L413: Hence, the overall net precipitation (precipitation-actual evapotranspiration) may change slightly across the climate models,

- L453-455: It is not clear to me how these numbers are derived from Figure 9. Thank you, no this is unclear. We have specified in the text: Even though it is not possible to completely separate the signal of these three components, a cautious estimation can again be achieved by subtracting the blue bars with the red bar result from LU0 in the reference period, so that the signal from the climate model bias is tried to be removed.

SECT. 5 (DISCUSSION AND CONCLUSIONS) - L480: replace 'the full range' by 'a range' as a limited number of scenarios were used, which cannot comprehend all possible futures. This has been corrected as suggested.

- L515-516 'that should be addressed along with other known sources of uncertainty such as climate model projections, land use projections and hydrological model structure uncertainty.': This discussion needs to be expanded. In particular, uncertainty in model parameter values can also affect the results. We have expanded this section L710-719: The indication that errors can be up to 10% is based on only a single case study with one catchment, one model and a limited number of land use and climate change scenarios. While similar results may be found when applying the same approach for catchments governed by the same dominant flow processes and land use types, like the one investigated in this study. The error must be expected to be site and context specific and therefore causes projection uncertainties that should be addressed along with other known sources of uncertainty such as climate model projections, land use projections, parameter uncertainties, geological uncertainty, and hydrological model structural uncertainty (Hansen et al., 2014; Karlsson et al., 2016). Furthermore, during calibration and setup of the model assumptions must be made, adding to the uncertainty. Parameter estimation are here done in a stepwise fashion, and the catchment scale calibration of Daisy along with a particle tracking approach limits the evaluation of performance of the nitrate component to mean catchment figures, and the dynamic of the nitrate system is thus impossible to verify. To account for this a full solute transport solution would be necessary but was unfortunately not possible in the framework of this study; but would be a relevant next step in investigating uncertainties and improve model verification. Hansen, A.L., Gunderman, D., He, X., Refsgaard, J.C., 2014. Uncertainty assessment of spatially distributed nitrate reduction potential in groundwater using multiple geological realizations. Journal of Hydrology, 519, Part A: 225-237. DOI:http://dx.doi.org/10.1016/j.jhydrol.2014.07.013 Karlsson, I.B. et al., 2016. Combined effects of climate models, hydrological model structures and land use scenarios on hydrological impacts of climate change. Journal of Hydrology, 535: 301-317. DOI:http://dx.doi.org/10.1016/j.jhydrol.2016.01.069

MINOR EDITS: - L143 'section 0': please add the correct section number. This has been added

- L170 and in the figure captions: add 'cell' after grid. Corrected

- L230 'no 1': do the authors refer to scenario 1 in Table 3? Please clarify. Yes, it refers to scenario 1. We have added more explanation: Baseline: The term baseline refers to results from the specific model run combination (scenario number1, Table 3), where current land use scenario (LU0) is combined with the observational climate data.

- L236: replace 'm2/s' by 'm3/s'. This has been corrected.

- L345: remove 'impact' after 'change'. This has been corrected.

- L348: there is something wrong here. Maybe 'different' need to be removed? Some words are missing, we have added: different due to

- L364-365: Please correct by 'the change in the drain flow fraction' (two occurrences). This is corrected

- L397 'this is also found for one of the models': remove 'also' (possibly replace by 'indeed'). This is corrected

REFERENCES: Knoll, L., Breuer, L., & Bach, M. (2020). Nation-wide estimation of groundwater redox conditions and nitrate concentrations through machine learning. Environmental Research Letters, 15(6). https://doi.org/10.1088/1748-9326/ab7d5c Tesoriero, A. J., Terziotti, S., & Abrams, D. B. (2015). Predicting Redox Conditions in Groundwater at a Regional Scale. Environmental Science and Technology, 49(16), 9657–9664. https://doi.org/10.1021/acs.est.5b01869 Wade, A. J., Jackson, B. M., & Butterfield, D. (2008). Over-parameterised, uncertain "mathematical marionettes" - How can we best use catchment water quality models? An example of an 80-year catchment-scale nutrient balance. Science of the Total Environment, 400, 52–74. https://doi.org/10.1016/j.scitotenv.2008.04.030

Please also note the supplement to this comment:
https://hess.copernicus.org/preprints/hess-2020-570/hess-2020-570-AC2-

supplement.pdf

---

## Author Comment (AC3) · 6 May 2021

Reply: Referee #3 General comments The authors investigate how nitrate reduction is affected by climate and land use changes for one Danish catchment. This is important as nitrate reduction maps are usually considered to be constant in time, which might not be appropriate for water quality management. With a modelling approach using MIKE SHE and Daisy, the authors show that climate has a stronger impact on nitrate reduction.

General comments: - The introduction is too brief; it should inform why land use and

climate changes are relevant to NO3-reduction. This is not explicitly mentioned. The key word "denitrification" also needs to be included. Thank you for pointing out this issue. We have included the denitrification term and added a paragraph on climate change/land use influences on reduction in L73-80: Even as, the link between climate change, land use change and nitrate reduction has been established in previous studies (e.g. Fleck et al., 2017; Mas-Pla and Menció, 2019; Olesen et al., 2019; Ortmeyer et al., 2021; Sjøeng et al., 2009). Ortmeyer et al. (2021) used a water balance model combined with a lumped-parameter nitrate mass model for an area in Germany, finding that nitrate concentrations in the groundwater increased towards the end of the century by up to 89 % as a result of changes in temperature, evapotranspiration and precipitation. Mas-Pla and Menció (2019) found that climate change in turn affects groundwater recharge and thus the dilution of nitrate in the subsurface in a study in Catalonia. While, Paradis et al. (2016) found that new agricultural practices under changing climate conditions led to substantial nitrate increases on an Island in eastern Canada.

Fleck, S. et al., 2017. Is Biomass Accumulation in Forests an Option to Prevent Climate Change Induced Increases in Nitrate Concentrations in the North German Lowland? , 8(6): 219. Mas-Pla, J., Menció, A., 2019. Groundwater nitrate pollution and climate change: learnings from a water balance-based analysis of several aquifers in a western Mediterranean region (Catalonia). Environ Sci Pollut Res Int, 26(3): 2184-2202. DOI:10.1007/s11356-018-1859-8 Olesen, J.E. et al., 2019. Nitrate leaching losses from two Baltic Sea catchments under scenarios of changes in land use, land management and climate. Ambio, 48(11): 1252-1263. DOI:10.1007/s13280-019-01254-2 Ortmeyer, F., Mas-Pla, J., Wohnlich, S., Banning, A., 2021. Forecasting nitrate evolution in an alluvial aquifer under distinct environmental and climate change scenarios (Lower Rhine Embayment, Germany). Science of The Total Environment, 768: 144463. DOI:https://doi.org/10.1016/j.scitotenv.2020.144463 Paradis, D. et al., 2016. Groundwater nitrate concentration evolution under climate change and agricultural adaptation scenarios: Prince Edward Island, Canada. Earth Syst. Dynam.,

7(1): 183-202. DOI:10.5194/esd-7-183-2016 Sjøeng, A.M.S., Kaste, Ø., Wright, R.F., 2009. Modelling future NO3 leaching from an upland headwater catchment in SW Norway using the MAGIC model: II. Simulation of future nitrate leaching given scenarios of climate change and nitrogen deposition. Hydrology Research, 40(2-3): 217-233. DOI:10.2166/nh.2009.068

- The methods should be more clearly described and structured. It is hard to follow the steps sometimes. Why are there three subsections on the nitrate model in the Method section? Is there potential to merge them? From reading the titles I do not directly know where to expect what content. Why do you define the terms in the very end of the chapter, not when talking about the data or modelling periods? Consider restructuring. Thank you for this comment. Yes, the section was unfortunately somewhat unclear. We have restructured and added extra information to the method section and changed the headlines, as well as added some extra introductory sentences to the method section (L121-L137). The reason for defining the period terms at the end of the chapter is due to the fact that the future scenarios have not yet been introduced in the Study site chapter, and that calibration/validation periods are not the same as the complete observational period.

- The evaluation of the model needs to be more in depth. The calibration approach needs to be better explained. Was there no calibration with nitrate concentrations? Validation results should be presented and model uncertainties in relation to the calibration/parameters need to be discussed. Maybe a sensitivity analysis would be helpful. The results are very long compared to a very short discussion and no conclusion section, consider streamlining and moving content to supplemental material. Thank you for pointing out this weakness. We have restructured and added text to the method section to improve the paper and added information and restructured the results section (4.1) on validation of the model. As suggested, we have moved figures to the supplementary material and added a conclusion (see response to L516).

- Please revise the consistency (e.g. N, nitrate, nitrogen usage (e.g. L124) or L 126)

and language (e.g. sometimes singular and plural are mixed or incomplete) Thank you, we have gone through the text to streamline definitions.

Specific comments: - L9-11 long first sentences, consider splitting. This sentence has been revised: Nitrate reduction maps have been used routinely in Northern Europe for calculating efficiency of remediation measures and impact of climate change on nitrate leaching. These maps are therefore valuable tools for policy analysis and mitigation targeting.

- L10: impact "of" climate change Corrected.

- L14: consider rewording "potential errors", what errors? This is unclear to me at this point Thank you, this sentence has been revised.

- L20: What do the authors propose to constrain the uncertainty of model formulation and assumptions? This is a valid and important question, but unfortunately not one that can be answered easily. One approach could be testing multiple model setups or types on the same study area, however due to computational and time limitations this is however not always feasible. We have added a few reflections on this in the abstract: To account for this uncertainty multiple approaches, assumptions and models could be applied for the same area, however as these models are very time consuming this is not always a feasible approach in practice. An uncertainty in the order of 10% on the reduction map may have major impacts on practical water management. It is therefore important to acknowledge if such errors are deemed acceptable in relation to the purpose and context of specific water management situations.

Introduction: - L35: "The amount of nitrate reduction occurring in groundwater depends on the flow paths and the depth to the redox interface." is very brief considering for example the Damköhler number. This part would benefit from a bit more in depth. What about availability of electron donors? We have added some more text on this: In the groundwater zone, nitrate reduction (nitrate reduction) takes place when nitrate containing water migrates from aerobic to anaerobic conditions and inherent reduced

compounds are available (Hansen et al., 2014; Postma et al., 1991). For quaternary sediments these reduced compounds are mainly organic carbon and pyrite and ferrous ion from clay minerals (Ernstsen and Mørup, 1992; Postma et al., 1991). This transition zone between aerobic and anaerobic conditions is denoted the redox interface.

Ernstsen, V., Mørup, S., 1992. Nitrate reduction in clayey till by Fe(II) in clay minerals. Hyperfine Interactions, 70(1): 1001-1004. DOI:10.1007/BF02397497 Hansen, A.L., Christensen, B.S.B., Ernstsen, V., He, X., Refsgaard, J.C., 2014. A concept for estimating depth of the redox interface for catchment-scale nitrate modelling in a till area in Denmark. Hydrogeology Journal, 22(7): 1639-1655. DOI:10.1007/s10040-014-1152-y Postma, D., Boesen, C., Kristiansen, H., Larsen, F., 1991. Nitrate Reduction in an Unconfined Sandy Aquifer: Water Chemistry, Reduction Processes, and Geochemical Modeling. Water Resources Research, 27(8): 2027-2045. DOI:10.1029/91wr00989

- L36: "In areas with Quaternary sediments characterized by groundwater dominated flow patterns and a relatively shallow redox interface, the N-reduction in groundwater can be the dominant removal process." Please provide a reference Thank you, a reference has been provided: In areas with Quaternary sediments characterized by groundwater dominated flow patterns and a relatively shallow redox interface, the nitrate reduction in groundwater can be the dominant removal process (Hansen et al., 2009).

Hansen, J.R. et al., 2009. An integrated and physically based nitrogen cycle catchment model. Hydrology Research, 40(4): 347-363.

- L43: I do not know why showing maps is considered as "a new approach". Merz et al. 2009, for example, also showed retention maps and NO3 half-life times. Yes, that is true. The sentence has been revised: An approach for utilizing and illustrating the results and the spatially varying nitrate removal fractions (percentages) are through a nitrate reduction map

– L47: "produced N-reduction maps with a 100 m spatial resolution for a 101 km2

catchment in Denmark," is not helping the argumentation, can be removed.

The sentence has been revised: Hansen et al. (2014) produced nitrate reduction maps for a 101 km2 catchment in Denmark, showing that nitrate reduction may vary from 20% to 70% between neighboring agricultural fields located only a couple of hundred meters apart.

Hansen, A.L., Christensen, B.S.B., Ernstsen, V., He, X., Refsgaard, J.C., 2014. A concept for estimating depth of the redox interface for catchment-scale nitrate modelling in a till area in Denmark. Hydrogeology Journal, 22(7): 1639-1655. DOI:10.1007/s10040-014-1152-y

- L50: This sentence has to be checked for grammar. It is also partly redundant with the next sentence. Thank you, this sentence has been removed.

- L55: "the effect hence is relatively large" please be more specific, "effect" is too vague here. We have specified that it is mitigation effects.

- L57: This sentence is not clear to me. You are saying that N-reduction maps can be used more easily than hydrological models, but actually those models are used to produce the maps. This contradicts. Thank you for this comment. This was indeed not very clear. We have revised the sentence in L68-L70: Using nitrate reduction maps based on a single model run, is clearly a much faster method than running multiple complex hydrological simulation models for large ensembles of scenarios and is therefore a practical tool for policy analysis

And a few places in the section above we have changed "nitrate maps" to singular form, to indicate more clearly that nitrate reduction maps are based on a single run.

- L60: Please revise the sentence. Thank you, this sentence has been revised: It is therefore very relevant to investigate the potential error arising when nitrate reduction maps are assumed to be constant in time. No studies have been reported on that issue.

Methods: - L74: "best", "long", "near-complete" Please, specify Thank you, the details of the timeseries have been specified: The discharge station at Kratholm has one of the best nutrient time series in Denmark starting in the 1980s, with near-daily sampling from 1989 (Windolf et al., 2016). The station, therefore, provides a long and near-complete data set for nutrient modelling as well as an extensive water discharge time series (Trolle et al., 2019). The average discharge amounts to 4.4m3/s and the load is approximately 14 kg NO3-N/ha/year.

Trolle, D. et al., 2019. Effects of changes in land use and climate on aquatic ecosystems: Coupling of models and decomposition of uncertainties. Science of The Total Environment, 657: 627-633. DOI:https://doi.org/10.1016/j.scitotenv.2018.12.055 Windolf, J. et al., 2016. Successful reduction of diffuse nitrogen emissions at catchment scale: example from the pilot River Odense, Denmark. Water science and technology : a journal of the International Association on Water Pollution Research, 73(11): 2583-9. DOI:10.2166/wst.2016.067

- L76: "The average discharge amounts to 4.4m3/s and the load is approximately 14 kg NO3-N/ha/year." Reference or more details needed Time period for the averages are added, as well as a reference on trends in the nitrate time series. See the comment above.

- L79: reformulate "There were measurements" Sentence has been revised: 226 measurements of the redox depth are available from boreholes in the area, the redox depths were mainly interpreted based on sediment colour as described by e.g. Ernstsen and Mørup (1992), and a few by measurements of reduced compounds.

Ernstsen, V., Mørup, S., 1992. Nitrate reduction in clayey till by Fe(II) in clay minerals. Hyperfine Interactions, 70(1): 1001-1004. DOI:10.1007/BF02397497

-L84: revise "100 meter redox depth map", I assume you mean the resolution Thank you, this has been corrected.

- L84: "This map" reference unclear, as citations mismatch, it is not clear to me. Thank you, this has been revised: A recent redox depth map of Denmark was created in 2019, where measurements and system variables were used in a machine learning environment to create a detailed redox depth map in 100 meter resolution. This newer map also indicates that the redox depth in the study area is predominantly shallow with 1-10 meter depth, and very few sites of 10-15 meters depth (Koch et al., 2019a; Koch et al., 2019b).

Koch, J., Ernstsen, V., Højberg, A.L., 2019a. Dybden til redoxgrænsen, 100 m grid. In: (GEUS), D.N.G.U.f.D.o.G. (Ed.), Copenhagen, GEUS. DOI:https://data.geus.dk/geusmap/?mapname=denmark#baslay=baseMapDa&optlay=&extent=-139925.69284407864,5929490.944444444,1254925.6928440786,6520509.055555556&layers=redox_dybde_100m_grid
Koch, J. et al., 2019b. Modeling Depth of the Redox Interface at High Resolution at National Scale Using Random Forest and Residual Gaussian Simulation. 55(2): 1451-1469. DOI:10.1029/2018wr023939

- L110: Please give a reference or indication why drains are needed, if locations are unknown. How do you define the "drain level" without this information? A more detailed explanation is now provided in L145-149: The study is heavily tile drained, and it can be assumed that drainage will always be present when it is need in the agricultural areas. However, the actual site-specific location of tile drains are unknown and therefore drains are specified across the entire catchment at a depth of -0.5 meters . Drain flow is however only activated when groundwater level rise above drain level. Apart from representing tile drainage the drainage system also represents small ditches and stream, too small to incorporate into the river system following the approach of Troldborg et al. (2010).

Troldborg, L. et al., 2010. DK-model2009 - Modelopstilling og kalibrering for Fyn. GEUS Report.

- L119 typo "percolation" Thank you, corrected.

-L119: "following methods" specify. Do you use several? Thank you. Yes, the section is very unclear. The complete section on Daisy modelling and calibration have been rewritten: The Daisy model setup for the Odense is contains on roughly 12,000 1D Daisy columns, and the water balance module is based on a previous calibration of the catchment (Børgesen et al., 2013), where root zone leaching and groundwater abstraction is compared with river discharge (Børgesen et al., 2013; Refsgaard et al., 2011). The model is setup so that each column represent unique combinations of soil type, climate, crop rotation and groundwater depth. The Daisy model uses the same climate input and soil parameter setup as MIKE SHE and the sensitive and calibrated unsaturated soil parameter from MIKE SHE were therefore transferred to Daisy. A more detailed description of the Daisy setup can be found in Karlsson et al. (2016). The water balance performance of Daisy was evaluated in the same calibration (2004-2007) and validation periods (2000-2003/2008-2009) as MIKE SHE.

Daisy also simulates nitrate leaching for each soil column that represents a unique combination of soil type, climate, crop rotation and groundwater depth. Crops are fertilized with mineral and organic nitrogen dependent on the farm type and soiltype. The crop recommended nitrogen rate for the years 2004-2007 was used to setup the fertilization scheme. Nitrate leaching input are simulated on daily basis based on the leaching from the permutated crop rotations simulated for the dominating soil type within a 200m x 200 m square grid (Karlsson et al., 2016). Because of the close feedback mechanism between nitrogen yields and nitrate leaching, the simulated mean nitrogen yields were recalibrated to observed annual mean nitrogen yields on Funen (Statistikbanken, 2015) for the dominating soil type for the period 2004-2007. The calibration is conducted by adjusting the crop parameters, following the methodology of Styczen et al. (2004). Nitrogen concentrations of yields were extracted from table values of mean nitrogen contents for different crops (Møller et al., 2005). For crop rotations including clover grass and peas nitrogen biological fixation is calculated using Høgh-Jensen et al. (2004) and nitrogen atmospheric deposition is included as input to the soil using standard Daisy settings for dry and wet deposition

(Hansen et al., 2012). Børgesen, C.D. et al., 2013. Udviklingen i kvælstofudvaskning of næringsstofoverskud fra dansk landbrug for perioden 2007-2011. Evaluering af implementerede virkemidler til reduktion af kvælstofudvaskning samt en fremskrivning af planlagte virkemidlers effekt frem til 2015, DCA - Nationalt Center for Fødevarer og Jordbrug, Tjele, Denmark. Hansen, S., Abrahamsen, P., T. Petersen, C., Styczen, M., 2012. Daisy: Model Use, Calibration, and Validation. Transactions of the ASABE, 55(4): 1317. DOI:https://doi.org/10.13031/2013.42244 Høgh-Jensen, H., Loges, R., Jørgensen, F.V., Vinther, F.P., Jensen, E.S., 2004. An empirical model for quantification of symbiotic nitrogen fixation in grass-clover mixtures. Agricultural Systems, 82(2): 181-194. DOI:https://doi.org/10.1016/j.agsy.2003.12.003 Karlsson, I.B. et al., 2016. Combined effects of climate models, hydrological model structures and land use scenarios on hydrological impacts of climate change. Journal of Hydrology, 535: 301-317. DOI:http://dx.doi.org/10.1016/j.jhydrol.2016.01.069 Møller, J. et al., 2005. Fodermiddeltabel - Sammensætning og foderværdi af fodermidler til kvæg. 64. Refsgaard, J.C. et al., 2011. Vandbalance i Danmark - Vejledning i opgørelse af vandbalance ud fra hydrologiske data for perioden 1990-2010, Copenhagen, Denmark. Statistikbanken, 2015. Statistical regional registrated annual mean yields. (In Danish) https://www.statistikbanken.dk/jord3. Styczen, M. et al., 2004. Standardopstillinger til Daisy-modellen. Vejledning og baggrund, Institut for Vand og Miljø, DHI.

- L121: when is the simulation period? Calibration was mentioned, but what about validation period? This has been added: The water balance performance of Daisy was evaluated in the same calibration (2004-2007) and validation periods (2000-2003/2008-2009) as MIKE SHE.

- L129: this is not a sentence The sentence has been corrected: Nitrogen concentrations of yields were extracted from table values of mean nitrogen contents for different crops

- L135: bad title The title has been changed to: 3.3 Calibration of the nitrate model (Phase 3)

- L139: "If the particle penetrates the redox interface, the nitrate is assumed to be removed completely and instantaneously by denitrification." Please, reflect more on this assumption. This is a generally accepted assumption in geochemistry (Postma et al., 1991; Hansen et al., (2014). We have added reference to validate this assumption. Hansen AL, Christensen BSB, Ernstsen V, He X, Refsgaard JC (2014) A concept for estimating depths of the redox interface for catchment scale nitrate modelling in a till area. Hydrogeology Journal, 22, 7, 1639-1655. Postma D, Boesen C, Kristiansen H, Larsen F (1991) Nitrate reduction in an unconfined sandy aquifer: water chemistry, reduction processes, and geochemical modeling. Water Resources Research 27:2027–2045.

- L143: section 0 Corrected to section 3.4

- L175-181: Please, specify how you can state that this was or was not the case? How did you further investigate the stuck particles? How many particles get stuck? It seems quite a lot if the correction causes changes between -7 and 9%. Yes, this issue is very frustrating. The issue has been observed before, when running Mike She with particle tracking. From the tracking file we can see that these particles do not leave their cell of origin. We have tried different approaches to eliminate the stuck particles (e.g. placement of the particles in different locations within the cell). Unfortunately, there was no fix for this issue, even as it was reported to the responsible company behind Mike She (DHI). We have estimated the number of stuck particles in the model an average of 9 % of particles.

- L183: Revise the sentences. Also, what was tried to improve the numerical difficulties? This sentence has been revised and more information added: Mike She is a commercial modelling tool and therefore there is no possibility to access the modelling code in order to correct this numerical error, or in any other way account for this model limitation. Therefore, it was necessary to introduce a correction scheme.

- L184: I did not understand how the correction was done and also why this approach

was used. Please, explain. We have added a better explanation on how this correction was done: The actual fate of these stuck particles (reduced/non-reduced) are unknown. At an early stage the assumption was made that the captured particles, if they had moved correctly through the system, would be subject to a fate similar to the non-captured particles, i.e. that the relationship between reduced/non-reduced was the same. If this assumption is valid the calculation the reduction potential in each grid cell is the same with/without the stuck particles. Unfortunately, this assumption may not always be valid. Furthermore, the arrival percentage estimated by the two methods are not the same as not all particles are released in the complex particle arrival count, the data from which is the only way to calibrate the nitrate model. For the two methods to be comparable it is therefore necessary to exclude the particles that are stuck in the unsaturated zone. The correction factor is therefore introduced to eliminate the particles that are stuck from changing the reduction map. The correction uses a simple linear equation, where a correction factor is manually fitted so that the arrival percentage (originating from the reduction map multiplied by the nitrate input) matches the particle arrival percentage. These corrections are done individually for all reduction maps, and the correction causes a change in the reduction in the range of -7% to 9% with a mean of 2%.

- L196: The reference seems quite old for climate projections. Thus projections for the end of this century might contain much higher uncertainty. Please explain why you used this one and not a newer study. Thank you for this comment, it is entirely true that the newer projection may be less uncertain. Unfortunately, the study was conducted some years ago and is therefore using the climate projections available at the time. We, unfortunately, do not have the possibility to rerun the complete modelling suite with the updated climate change scenarios.

- L197: I do not know why bias-corrections are necessary, please explain. Thank you for your question. The bias correction is done because the regional climate model operates on much larger scale than hydrological models. Local precipitation patterns

are thus not necessarily correct represented in the regional models. Bias correction to local observations is therefore a common approach for dealing with this issue e.g., Chen et al. (2011), Pasten-Zapata et al., (2019); Refsgaard et al., (2016). We have added the word downscaled to the sentence to help.

Chen, J., Brissette, F.P., Leconte, R., 2011. Uncertainty of downscaling method in quantifying the impact of climate change on hydrology. Journal of Hydrology, 401(3–4): 190-202. DOI:http://dx.doi.org/10.1016/j.jhydrol.2011.02.020

Pasten-Zapata, E., Sonnenborg, T.O., Refsgaard, J.C., 2019. Climate change: Sources of uncertainty in precipitation and temperature projections for Denmark. Geological Survey of Denmark and Greenland Bulletin, 43: e2019430102-01-e2019430102-06. DOI:https://doi.org/10.34194/GEUSB-201943-01-02

Refsgaard, J.C. et al., 2016. Climate change impacts on groundwater hydrology – where are the main uncertainties and can they be reduced? Hydrological Sciences Journal, 61(13): 2312-2324. DOI:10.1080/02626667.2015.1131899

- L200: a "combined" median model? Thank you, this has been explained better: The four selected realizations represent a wet, +19% in precipitation (ECHAM-HIRHAM5), a dry, -11% decrease in precipitation (ARPEGE—RM5.1), a warm, +3.4 °C temperature increase (HadCM3-HadRM3) and a model representing a median projection, +10% in precipitation and +2.1 °C in temperature (ECHAM5-RCA3).

- L210 I do not understand "3% point reduction". What is point telling here? We wanted to indicate that it is percentage point difference and not a difference in percentage. We have change it to p.p. which may be a more common way to write it.

- L218f.: I do not understand this combination that was done: "compared to the climate model results found for the reference period 1990-2009 using the same land use scenarios, resulting in 32 scenarios." I think formulations are overcomplicated and Table 3 should be placed here. Thank you for this comment. We have moved table 3 to this

location and revised the sentence: All 20 combinations of future climate projections (4) and land use (5) were specified as input to the hydrological model. The model was run for both future (2088-2099) and reference period (1990-2009) resulting in 40 scenarios. Additionally, the model was run with observed climate for the period 1990-2009 (5 scenarios using observed land use and the four land use scenarios).

Results: - L240: "observed trends in nitrate yields" where are these trends shown? Yes, trends is the wrong word. We have changed it to: observed values of nitrate yields

- Table 3: I think this should be presented in the Methods? 3.6 scenarios? What does the grey shade mean? How can the climate scenarios be used for 1990-2009? Or is it necessary? I do not see them later in the presented maps. Thank you, the table has been moved accordingly. The grey shades indicate the scenarios displayed in the coming figures 3,4,6,7,8, and the baseline scenario. We have added text in the figure caption to explain.

Thank you for you question on the climate scenarios. Climate models are commonly run for both a reference period in the present time and a future period. As climate model results are affected by biases, even though they are bias corrected/downscaled, the way of dealing with this issue is mainly solved by displaying only changes from past to future. This is done under the assumption that the climate model biases are constant (the same for present and future conditions), thereby changes in the impact results are deemed more trustworthy than actual values from the future projections. This is the reason why figure 4-9 all focus on the changes from past to future. We have added an extra explanatory sentence in line 312: Future climate model runs are always compared with results from this period for the relevant climate model to ensure that climate model biases do not dominate the results.

- Table 4: Do you have an idea why the standard deviations of all models are that similar (Table 4, 0.36-0.39)? Can you comment on that, please? This could be related to the fact that most changes in the reduction map are happening for values close or

around the mean. Areas with 0% og 100% reduction are perhaps less likely to change reduction potential, as they are either very close to surface water (0% reduction) or located at areas with a deep groundwater level (long deep flow paths and limited drain flow) (100% reduction).

- L295: "To investigate to what degree land use changes and climate change affect the reduction map, the difference between these scenarios and the reference scenario is shown in Figure 4." Does not seem to fit here if the next section title is "4.4 Impact of land use change on reduction maps". Yes this is confusing. We have moved this sentence and incorporated it into L391: To investigate the impact of land use change on the reduction maps, only land use is changed while climate remains constant, shown as the difference between land use changes scenarios and the baseline scenario (Figure 4, top row).

- Figure 3 and Figure 4 seem a bit redundant to me, considering that 4 is just the difference between the map shown in Fig2 and Fig3. I think one Figure could be economized here by merging or moving to the supplements Thank you for this suggestion, we have moved Fig. 3 to supplementary material, and moved the sentence in L274-276 connected to the figure with it.

- I think it is not necessary to show Fig7 and Fig8, especially because Fig.8 is mainly a reprint of Fig.12 in Karlsson et al. 2016, while the maps in Fig.7 do not allow to recognize more details than the general observation of land use changing little, two climate scenarios becoming wetter and two drier, which is also clear from Fig.5. Again I think redundancy should be reduced and plots removed or to the supplements. Thank you for this suggestion, we have moved Fig. 7 and 8 to supplementary material.

Discussion/Conclusions: - L510: "such effects" reference unclear. Please explain further how 10% change in nitrate reduction over almost a century relates to the uncertainties of nitrate reduction maps. Is it really essential to consider changes in nitrate reduction for management, if the tool itself is already quite uncertain? Thank you,

we have made this sentence more clearly: The uncertainty of using a fixed reduction map for future scenarios should of course be seen in the context of the inherent uncertainties of the nitrate reduction maps (Hansen et al., 2014b). We have also added some reflections on the impact for management, L593-595: 10% error on the reduction map may potentially have major impacts on practical water management. Considering for instance the baseline scenario in Table 4, where the average N-reductions vary between 55% and 67% reduction, this implies that the net impact of a 100 kg N reduction in leaching from the root zone will vary between 45 kg and 33 kg (i.e. 30%). Such changes are larger than the effects of sophisticated mitigation measures (Hansen et al., 2017). Hansen, A.L., Gunderman, D., He, X., Refsgaard, J.C., 2014b. Uncertainty assessment of spatially distributed nitrate reduction potential in groundwater using multiple geological realizations. Journal of Hydrology, 519, Part A: 225-237. DOI:http://dx.doi.org/10.1016/j.jhydrol.2014.07.013 Hansen AL, Refsgaard JC, Olesen JE, Børgesen CD (2017) Potential benefits of a spatially targeted regulation based on detailed N-reduction maps to decrease N-load from agriculture in a small groundwater dominated catchment. Science of the Total Environment, 595, 325-336.

- L512-516: "single case study" how representative is this case? What do you expect for other sites? Compare to other studies. General spatial differences between nitrate reduction could be considered. Thank you for this valid point. Unfortunately, we are not aware of any other studies evaluating uncertainties from application of a fixed reduction map vs. a full nitrate modelling scenario estimation. We have added a few remarks: The indication that errors can be up to 10% is based on only a single case study with one catchment, one model and a limited number of land use and climate change scenarios. While similar results may be found when applying the same approach for catchments governed by the same dominant flow processes and land use types, like the one investigated in this study. The error must be expected to be site and context specific and therefore causes . . .

- L513-516: You mention uncertainties of input data (climate and land use) and model

structure (though very briefly) but do not discuss the uncertainties related to the model and its parameters. Thank you, yes this is not mentioned. We have added text on this issue in L606-615: The error must be expected to be site and context specific and therefore causes projection uncertainties that should be addressed along with other known sources of uncertainty such as climate model projections, land use projections, parameter uncertainties, geological uncertainty, and hydrological model structural uncertainty (Hansen et al., 2014b; Karlsson et al., 2016). Furthermore, during calibration and setup of the model assumptions must be made, adding to the uncertainty. Parameter estimation are here done in a stepwise fashion, and the catchment scale calibration of Daisy along with a particle tracking approach limits the evaluation of performance of the nitrate component to mean catchment figures, and the dynamic of the nitrate system is thus impossible to verify. To account for this a full solute transport solution would be necessary but was unfortunately not possible in the framework of this study; but would be a relevant next step in investigating uncertainties and improve model verification.

- L516: This is not a nice ending. I would collect the conclusions in an extra Conclusion section. Thank you for this suggestion. We have added a conclusion section: Nitrate reduction maps are valuable tools used for calculation of remediation and climate change effects on nitrate leaching, and are generally considered constant in time, even though the timing of nitrate leaching, and flow paths may change. In this study we investigate the potential consequence for estimation nitrate climate and land use change impact projections when assuming a fixed reduction map. For an agricultural dominated catchment in Denmark, the Daisy model was used to provide nitrate leaching input, while the hydrological model Mike She was used to simulate the flow regime and nitrate flow path through particle tracking. Four land use scenarios and four climate change projections were evaluated. The main finding of the study was: • Changing climate conditions lead to reduction map changes of up to 10%; whilst effects from land use changes where minor. However, land use effects may be underestimated due to drainage formulations in non-agricultural areas. • Thus, the uncertainty of the

reduction maps is dependent on both model setup and assumptions, the catchment flow regime as well as affected by the span of the chosen land use and climate change scenarios. â˘Ać The error will therefore be specific for the study site and context and it should, consequently, be tackled along with other sources of uncertainty, like geological, parameter, and model structure uncertainties that are not evaluated in this study.

Data/code availability: - "owned by the DMI" - what does this mean? Where and how to access it? The data is currently not pubplily available from the Meteorological service, but will be in the future. We have added this information in L576-569: An exception to this is the observed climate data input which is currently the property of the Danish Meteorological Institute (DMI), but will be made publicly available through https://www.dmi.dk/frie-data/ before end of 2023.

Figures: - F1: typo at "Market driven". Odd start of the caption as the Figure is showing the study area and land use scenarios and not a "red square". Thank you, we have corrected the typo and the formulation in the figure text.

- F2: I think the map titles should be linked to the legend color bars. Maybe increase letter size of legend, add unit at left panel. To me it is unclear, what the text on the y axis refers to "Observed climate. . .", it seems unconnected. In the caption specify what the reduction refers to e.g. from. . . to. . .. Thank you for these suggestions. We have changed the figure according to the comments.

- F5: I suggest to remove redundant subplot titles, this should be explained in the caption. It has been removed

- F9: The caption is unclear to me. What combinations are shown? What is meant here "or the reduction map from the scenarios"? We have tried to explain better what we mean: Bars denote the change in nitrate flux at the catchment outlet that arises from using either a fixed nitrate reduction potential map (baseline) or using a reduction map based on the individual scenarios.

References: - There seems to be an error in the display of the references – is it double or some other problem? Yes, it is double, sorry. This has been corrected.

Please also note the supplement to this comment:
https://hess.copernicus.org/preprints/hess-2020-570/hess-2020-570-AC3-supplement.pdf

---

## Author Response (AR3)

**Report #1**

Suggestions for revision or reasons for rejection (will be published if the paper is accepted for final publication)

In the revised version of the manuscript, the authors clarified the methods used in Section 3 and added a discussion about other sources of uncertainty beyond climate and land use in Section 5.2. The analyses performed and the results remain unchanged. I think that the manuscript has significantly improved in terms of clarity. However, I still have a number of comments, in particular regarding the nitrogen and climatic input data and the derivation of the value of 10% for the uncertainty in the reduction map. In addition, although the authors did not assess the uncertainty coming from the model parameters, the manuscript highlights other sources of uncertainty (bug in Mike She code or model error in validation) that are of the same order of magnitude as the uncertainty due to climatic/land use conditions (10%). I think that this should be better discussed.

Importantly, I would like to make a clarification with respect to the authors' reply to my second comment that 'A global search engine (Shuffled Evolution Complex) was used. Thus, limiting the risk of equifinality'. I would like to highlight that this statement is not correct. Equifinality results from a lack of information/data to constrain the parameters of a given model, and does not depend on the method used to estimate the parameters. Optimization method such as the Shuffled Evolution Complex algorithm may not reveal equifinality because they identify only one parameter set that 'best' fit the data, while there is likely to be other parameter sets with similar performance that have very different parameter values. I refer e.g. to Beven et al. (2006) for a discussion on the issue of equifinality.

Thank you for this comment. Yes, you are of course completely right. We meant to say that the risk of landing in a local minimum was reduced by the global search engine. This is not equifinality, this was entirely a mistake from our side.

We have made the discussion on the uncertainty clearer, e.g., by introducing a new section (5.3), where all reflections on uncertainties and limitations are gathered from the other parts of the discussion. We have also added and modified sections on the above issue on split-sample and equifinality in L645-655 and L679-692:

*L645-655: When using a hydrological model for simulating impacts of changes in catchment conditions compared to those existing in the calibration period, split sample validation tests are not sufficient to document a model's capability to simulate hydrological changes. Experience shows that models that are used for making predictions beyond the conditions for which they are calibrated, such as land use or climate change in the present study, due to equifinality often suffer from model structural uncertainties (Refsgaard et al., 2012). In such situations the more comprehensive and data demanding differential split sample tests are recommended (Klemeš, 1986; Refsgaard et al., 2014). Due to lack of data such tests were beyond the scope of the present study. Instead, the model structural uncertainty for the present case was assessed using a multi-model approach with two additional hydrological models (Karlsson et al., 2016) suggesting that the signal coming from climate change was dominating over model structural uncertainty as far as hydrological change is concerned. Therefore, we argue that the inevitable uncertainties arising from model use beyond calibration conditions most likely are not so large that they affect our conclusions.*

*L679- 692: During calibration and validation of the model, a decrease in model performance was registered in the water balance for the validation period. This could be caused by non-optimal parameter estimates, and there is always a risk of equifinality during model calibration. In this case, the risk was minimized using*

*an extensive dataset of both discharge, hydraulic head, redox depth and nitrate flux during calibration of the different model steps. However, a multi model set as used in this study may still be prone to the risk of equifinality. Parameter estimation are here done in a stepwise fashion for each of the models, and the catchment scale calibration of DAISY along with the particle tracking approach limits the evaluation of performance of the nitrate component to mean catchment values. The dynamic of the nitrate system is thus impossible to verify. To account for this a full solute transport solution would be necessary but was unfortunately not possible in the framework of this study; but would be a relevant next step in investigating uncertainties and improve model verification.*

*During simulations it was found that some particles were not correctly released due to model error, and we therefore were forced to make the assumption that the particles would be distributed similar to non-trapped particles. However, the validity of this assumption is associated with considerably uncertainty. The correction led to mean changes in reduction of 2 %, but the resulting impact on the true reduction map is unknown, as we do not know the actual travel path of the trapped particles.*

Klemeš, V., 1986. Operational testing of hydrological simulation models. Hydrological Sciences Journal, 31(1): 13-24. DOI:10.1080/02626668609491024

Refsgaard, J.C. et al., 2012. Review of strategies for handling geological uncertainty in groundwater flow and transport modeling. Advances in Water Resources, 36: 36-50. DOI:https://doi.org/10.1016/j.advwatres.2011.04.006

Refsgaard, J.C. et al., 2014. A framework for testing the ability of models to project climate change and its impacts. Climatic Change, 122(1): 271-282. DOI:10.1007/s10584-013-0990-2

I report below detailed comments.

p9 L203: Could the authors briefly explain what the crop recommended nitrogen rate is and where the value come from (reference)?

Thank you, we have added the following statement: *The crop recommended nitrogen rate based on soil type and crop sequence from Danish Ministry on Agriculture (Plantedirektoratet, 2005) for the years 2004-2007 was used to setup the fertilization scheme.*

p9 L210: Could the authors add the source of the nitrogen atmospheric deposition dataset.

Thank you for this comment. The source is given at the end. We have tried to indicate this better by moving the reference: *For crop rotations including clover grass and peas nitrogen biological fixation is calculated using Høgh-Jensen et al. (2004) and nitrogen atmospheric deposition is included as input to the soil using standard DAISY settings (given in Hansen et al., 2012b) for dry and wet deposition.*

p12 L279: The calculation of potential evapotranspiration is still unclear. What is 'water vapour' and 'water pressure'? I guess that the authors mean vapour pressure/relative humidity and atmospheric pressure, which are the variables used in the Penman Monteith equation. In addition, more details are required on the radiation term. Net radiation is necessary to compute Penman Monteith equation, while typically climate models provide the downward radiation terms only.

Thank you for pointing out this shortcoming. We have changed the sentence to: *The reference evapotranspiration is calculated using FAO Penman– Monteith formula adapted by Allen et al. (1998) based on the climate model outputs for minimum and maximum temperature, incoming long and short wave solar*

*radiation, relative humidity and wind speed. Following the recommendations in Allen et al. (1998) and Seaby et al. (2013) variables needed for the Penman– Monteith formula e.g., net radiation (calculated from the net incoming short and long wave radiation), water vapour pressure, height-adjusted wind speed and atmospheric pressure, where calculated from these outputs.*

p18 Table 4: I guess that the mean and standard deviation refer to the spatial distribution of the nitrate distribution potential? Please clarify.

We are not totally sure what is meant, but we have tried to make the table text of what is shown clearer: *Table 4: Mean and standard deviation (in brackets) across the catchment for each of the nitrate reduction potential maps (proportion of nitrate N reduced).*

p21 L421: Figure 4 does not report results on reduction potential. Please refer to the appropriate figure/table for this.

True this should be Figure 3. This has been corrected.

p28 L573-579: What about nitrate reduction in soils?

Yes, that is correct, we have not mentioned that the soil type could of course also influence the results. We have added a sentence on this in L657: *The present study was carried out for a groundwater dominated catchment characterized by till deposits, confined aquifers, and relatively shallow redox interfaces and phreatic groundwater tables. Furthermore, this catchment has a relatively uniform soil type distribution, dominated by clayey soils.*

p28 L594: Where does the value of 10% comes from? It is not clear from the results presented in Section 4 (Figure 6).

We have added a line on how this value is found in L570*: Across all climate models, the average absolute effect on nitrate arrival is 10%, when using a fixed reduction map compared to a targeted reduction map.*

p29 L602-603: I think that this uncertainty of up to 10% should be also discussed with respect to other modelling uncertainties that are reported in the manuscript and that are of the same order of magnitude, i.e. the uncertainty due to the bug in the Mike-SHE code (that is up to 9% p12 L267), and the model error in the validation period (that is up to 10% for Mike She p15 L328 and up to 23% for Daisy model p15 L333)

Thank you for this comment. Please see the reply for the main comment above.

p29 L612: There is something wrong is this sentence. Please clarify.

This sentence has been deleted and the section moved to L666.

P30 L628 'Thus, the uncertainty of the reduction maps is dependent on both model setup and assumptions': to which model setup and assumptions do the authors refer? The uncertainty to model setup and assumptions is not analysed in the study, but only discussed as potential uncertainty that may arise. This second conclusion should be amended to reflect this.

Thank you for this comment, we have changed the sentence according to the suggestion: *The magnitude of the changes in the reduction map found here may, however, be influenced by both model setup (e.g., drainage), model errors (e.g., particle flow paths) and assumptions (e.g., fixed redox interface). Furthermore, the span of the chosen land use and climate change scenarios analysed and the flow regime in the study catchment may also influence results.*

Minor edits

p3 L57 'have been applied': what is the subject of 'have'? Is it 'a nitrate reduction map'? In this case it should be replaced by 'has'.

Thank you, this has been corrected

p4 L93: replace 'constitute' by 'constitutes'

Thank you, this has been corrected

p4 L94 'fellow': Do the authors mean 'fallow'?

Yes, this has been corrected

p4 L97: replace 'as a results' by 'as a result'

This has been corrected

p5 L112: replace 'in 100 meter' by 'at 100 meter'

This has been corrected

p6 L131: 'inverse' should be replace by 'inversely', but I would actually remove this term.

Thank you, this has been removed

p7 L147: replace 'need' by 'needed'.

This has been corrected

p10 L224'accumulated': do the authors mean 'cumulative'?

Yes, this has been corrected

p11 L239: replace 'representable' by 'representative'.

This has been corrected

p11 L261: add 'it' after ('the data from which')

This has been corrected

p14-15 Table 2: I suggest to refer to 'observational period' and 'reference period' rather than 'control period'

Thank you, this has been corrected

p16 Table 3 caption: replace 'for the crop type' by 'for the crop types'

This has been corrected

p20 L404-405: the verb is missing is this sentence.

Verb added

p22 L437: remove the 's' at the end of 'results' (in 'these changes results').

This has been corrected

'p.p.' is used multiple times in the manuscript: what does it stand for?

Thank you, yes we have not explained this. P.p. stand for percentage point, we have added the definition in Line 298

p25 L520: replace 'figure 9' by 'figure 6'.

This has been corrected

p30 L624-625: replace 'the main finding of the study was by 'the main findings of the study were'

This has been corrected

References

Beven, K. (2006). A manifesto for the equifinality thesis. Journal of Hydrology, 320(1–2), 18–36. https://doi.org/10.1016/j.jhydrol.2005.07.007

**Report #2**

Suggestions for revision or reasons for rejection (will be published if the paper is accepted for final publication)

The authors have addressed the previous comments and have greatly improved their manuscript and its clarity. I have a few more, but minor comments that should be addressed before publication of this manuscript.

General (minor) comments:

- I appreciate the much clearer Method Sections that you greatly improved. Just, sometimes I feel it is not completely which methods were set up previously in other studies and simply reused for this manuscript or which ones you extended from previous studies. One example is the part on redox depth estimation paragraph L236-242. Maybe in some cases it would be helpful to start with the reference instead of at the end of the description. "Following …, we did…" or "We extended on the method … presented by …." Please revise those sections to be clear. This is not easy as your work makes use of many previous approaches from the study region and in general, but I think it would further improve the readability and the connection to the references.

Thank you for this suggestion. We have tried to indicate this clearer in the method section in L147, L171, L201, L213, L232.

- Check consistency of spelling:

1.) NO3-N should all be NO3-N or "nitrate-N" ;

Thank you, this has been corrected.

2.) lower and upper case usage for each of the models MIKE SHE and DAISY;

Yes, this is inconsistent. We have corrected this to upper case for both models throughout the manuscript.

3.) When going through the manuscript I found mostly 2080-2099 for the future period, but in some instances 2088-2099, please revise to ensure these are correct.

Thank you, this has been corrected.

Detailed comments:

L 10: I still think this formulation is overly complicated. I suggest formulating the goal of your study in a positive way, something like investigating the "potential improvement of using transient nitrate reduction maps compared to …".

Thank you for this suggestion, we have added this in the text.

L 36: This sentence should be more precise in my opinion. Denitrification is one of the removal processes not just a synonym of the term, e.g. what about N uptake? Especially when you talk about processes in the unsaturated zone as you simulated with Daisy and in surface waters as you mention here, I think the terms should be introduced more carefully. It merits this attention for the aim of your study. I think the next paragraph would also fit to include "denitrification" term.

Thank you for this suggestion. We have amended the text to make the description more precise: *Nitrate is removed by a set of natural near-surface removal processes including plant uptake and soil retention,*

*furthermore, the natural removal of nitrate in the groundwater and the surface water must also be considered, when assessing the impacts of nitrate leaching from agricultural areas on aquatic ecosystems. This removal, takes place via natural biogeochemical reduction processes often referred to as denitrification It can be expressed as a percentage removal and depending on the actual hydrobiogeochemical conditions, the denitrification may mainly occur in groundwater or in surface water systems such as lakes or wetlands (Huno et al., 2018; Quick et al., 2019).*

L 40: remove "(nitrate reduction)"

This has been corrected.

L 42 and 45: inconsistent spelling lower and upper case "quaternary"

This has been corrected.

L 55: An approach is singular, please revise

This has been corrected.

L 102: "the load is approximately …" What kind of load? Is this input as N surplus or the export in the stream? This should be more specific, and depending on your answer possibly also include how it is estimated, e.g. mean discharge times mean concentration?

Thank you, we have modified the sentence to: *The station, therefore, provides a long and near-complete data set for nutrient modelling as well as an extensive water discharge time series (Trolle et al., 2019). In 2005-2009, the average discharge amounts to 4.6m³/s and the transport in the stream (load) is approximately 14 kg NO3-N/ha/year, calculated from measurements of mean concentration and mean water discharge.*

L 106: To start a sentence with a number is not a nice style. Plase revise

This has been corrected.

L125: Danish

This has been corrected.

L 155: I am curious: why did you select these periods for calibration and validation? Why do you use more years for validation (in total) than for calibration? From my knowledge this is rather uncommon, therefore I am curious about this decision.

The calibration period was chosen because this period has the best dataset, the first ten years (1990-2000) where used as warm-up. Subsequently, the validation period was then simply defined before and after the calibration period.

L231-235: Can these graphs be shown in the supplements? I think this could be good additional information. Also add reference to SI here.

Thank you for this suggestion, we have added the figure to the supplementary material and included a reference to the figure in the supplementary material in the text.

L 270: 16 models, but L 306 20 simulations. Is this correct?

Yes, this is correct. In the 20 simulations, there are also included the 5 runs with baseline land use. We have deleted this sentence as it is not necessary and causes confusion.

L 597: New inserts "N reduction" are not consistent with "nitrate reduction" in the rest of the text, recheck please

Thank you, this has been corrected.

L 615: "To account for this, a full solute" comma missing

Comma was added

Table 4: response to your response: "Table 4: Do you have an idea why the standard deviations of all models are that similar (Table 4, 0.36-0.39)? Can you comment on that, please?" – "This could be related to the fact that most changes in the reduction map are happening for values close or around the mean. Areas with 0% og 100% reduction are perhaps less likely to change reduction potential, as they are either very close to surface water (0% reduction) or located at areas with a deep groundwater level (long deep flow paths and limited drain flow) (100% reduction)." Maybe, that is what you meant. Actually, now with your response, I think that it is because all models cover the range of 0-100% with relatively high amount of cells in the extremes, stretching the standard deviation to similar values independent of the mean. I am not sure, the standard deviation is telling considering this perspective and the not normal distributed values. Figure 2b for example clearly shows that <10% and >90% prevail.

This sounds like a very valid reason for the similarity in the standard deviations. I think you are probably right, and therefore the standard deviations are maybe not so informative. We have chosen to leave them in, as it is difficult to test, if that is the reason in the actual case.

Report #3

Suggestions for revision or reasons for rejection (will be published if the paper is accepted for final publication)

The authors have made many changes in the revised version in an effort to address the comments from three reviewers. Unfortunately, they could not really address our concerns about equifinality, model calibration and uncertainty assessment, and perform a convincing evaluation of the model with observational data. The calibration of a parameter-rich model still relies on manual calibration of some parameters and with little use of observational data to evaluate the model. The research question "how changes in flowpaths and timing of nitrate release will influence denitrification in a context of global changes" is interesting, but the methods are relatively poor (although probably labour intensive).

While we agree that our model is parameter rich, we do not agree to the reviewer's statement that we have made "little use of observational data to evaluate the model". The reviewer statement about "concerns about equifinality, model calibration and uncertainty assessment" is very general without mentioning specific issues or problems. In the first round review, the reviewer expressed only two specific concerns (poor evaluation of interannual variability and need for differential tests related to land use and climate change) which we addressed in the discussion at that time. Altogether, we cannot follow the general statements expressed in the review, and we do not find the criticism of inadequate information on model calibration and evaluation fair, because:

- The modelling system used in the present study comprises three components: 1) The MIKE SHE hydrological model; 2) the DAISY root zone/nitrate model; and 3) the model describing the depth to the redox interface. Calibration and evaluation of the first two models (MIKE SHE and DAISY) was reported in a previously, well cited (92 ISI citations) journal paper (Karlsson et al., 2016). We do not think it is useful to repeat too much of the descriptions from the old paper, which we instead have referred to.
- For the hydrological model we used observational data from 415 groundwater wells and four discharge stations as calibration targets distributed across the 486 $km^2$ catchment. This is in our experience a relatively data-rich catchment suitable for the kind of analysis we have performed.
- It is correct that the hydrological model contains many parameters. However, through a rigorous sensitivity analysis including 43 parameters we ended up calibrating only the five most sensitive parameters. We used a global search parameter optimization algorithm reducing the risk of finding local optima. We believe that the hydrological model calibration and evaluation as reported in Karlsson et al. (2016) have been performed using state-of-the-art methodologies. Having said that we agree to the reviewer statement in the first review that it would have been useful with some kind of differential split-sample tests to evaluate the model's capability to simulate changes in land use and climate. As explained in the previous interactive discussions, such tests were not possible with the existing data. Instead the uncertainties in predictions of land use and climate change have been assessed using a multi-modelling approach in Karlsson et al. (2016). We agree that this issue has not been adequately addressed in the previous manuscript and we have therefore added a discussion on this concern and its possible implications for the study conclusions in our revised manuscript.
- The calibration of DAISY has been performed using the standard protocols for this type of model (Styczen et al., 2004) as described in Karlsson et al., 2016).

- The third model component, which was not included in Karlsson et al. (2016) is the model used to calculate the depth to the redox interface. As explained in the manuscript, this is a relatively simple model with only two parameters used for calibration against total N-load at the catchment outlet. To reduce the equifinality, the final values of the two parameters were fixed by comparing the calibrated redox interface to independent data on depth to redox interface collected from 226 wells. The information from these wells come from the national database that according to legislation stores information on all water supply wells in the country. According to our experience 226 wells with redox information within a 486 $km^2$ catchment is a quite high data density, and we cannot see how we possibly could have obtained better data or methods for evaluating the simulated depth to redox interface.

Based on the above we would argue that our modelling methodology basically is sound and suitable for the purpose of the study. We can agree to two of the specific concerns and have modified our discussion in the revised manuscript with respect to:

- We agree that some kind of differential split-sample test would have been useful to evaluate how good the model is in simulating effects of changes in land use and climate. But we argue that this aspect has been adequately evaluated by the multi-modelling study performed by Karlsson et al. (2016).
- We agree that our set of models inevitably includes some equifinality.

---

## Author Response (AR4)

**Reply Report #1**

The authors have addressed my comments. In particular, they now discuss the uncertainties and limitations of the study in a separate section (Sect. 5.3).

I just have one remaining comment regarding the sentence p29 L601-603 which needs to be reformulated. I cannot make sense of the end of the sentence 'due to equifinality often suffer from model structural uncertainties'. Models that have been calibrated over a given time period may fail to make predictions under different conditions because of equifinality, as stated by the authors, but also because model structures and parameter values may not be tranferrable in time (see e.g. Thirel et al., 2015).

References:

Guillaume Thirel, Vazken Andréassian & Charles Perrin (2015) On the need to test hydrological models under changing conditions, Hydrological Sciences Journal, 60:7-8, 1165-1173, DOI: 10.1080/02626667.2015.1050027

Thank you for pointing out this confusing paragraph, we have amended the sentence and added this very relevant reference to the paper:

*Experience shows that models that are used for making predictions beyond conditions for which they are calibrated (such as land use or climate change in the present study), often suffer from model structural uncertainties (Refsgaard et al., 2012), equifinality or that parameters may not be transferrable in time (Thirel et al., 2015).*